# LSH Tells You What To Discard: An Adaptive Locality-Sensitive Strategy for KV Cache Compression

## Abstract

Transformer-based large language models (LLMs) use the key-value (KV) cache to significantly accelerate inference by storing the key and value embeddings of past tokens. However, this cache consumes significant GPU memory. In this work, we introduce LSH-E, an algorithm that uses locality-sensitive hashing (LSH) to compress the KV cache. LSH-E quickly locates tokens in the cache that are cosine dissimilar to the current query token. This is achieved by computing the Hamming distance between binarized Gaussian projections of the current token query and cached token keys, with a projection length much smaller than the embedding dimension. We maintain a lightweight binary structure in GPU memory to facilitate these calculations. Unlike existing compression strategies that compute attention to determine token retention, LSH-E makes these decisions pre-attention, thereby reducing computational costs. Additionally, LSH-E is dynamic – at every decoding step, the key and value of the current token replace the embeddings of a token expected to produce the lowest attention score. We demonstrate that LSH-E can compress the KV cache by 30%-70% while maintaining high performance across reasoning, multiple-choice, long-context retrieval and summarization tasks.

## 1 Introduction

The advent of large language models (LLMs) has enabled sharp improvements over innumerable downstream natural language processing (NLP) tasks, such as summarization and dialogue generation (Zhao et al., 2023; Wei et al., 2022). The hallmark feature of LLMs, the attention module (Bahdanau, 2014; Luong, 2015; Vaswani, 2017), enables contextual processing over sequences of tokens. To avoid repeated dot products over key and value embeddings of tokens, a key-value (KV) cache is maintained in VRAM to maintain these calculations. This technique is particularly popular with decoder LLMs.

However, the size of the KV cache scales quadratically with sequence length $n$ and linearly with the number of attention layers and heads. Assuming the size of the KV cache is $n$ tokens, for each new decoded token, $n$ attention scores need to be added which requires a total of $\mathcal{O}(dn^2)$ computation, where $d$ is the projection dimension, and $\mathcal{O}(n^2)$ storage. For example, maintaining the KV cache for a sequence of 4K tokens in half-precision (FP16) can require approximately $\sim$16GB of memory for most models within the Llama 3 family (Dubey et al., 2024). These memory costs are exacerbated with batched inference and result in high decoding latency (Fu, 2024). Consequently, there is significant interest in compressing the size of the KV cache to enable longer context windows and low-resource, on-device deployment.

An emerging strategy for reducing the size of the KV cache is *token eviction*. This approach drops the key and value embeddings for past tokens in the cache, skipping future attention calculations involving these tokens. Various token eviction/retention policies have been explored in recent literature, including the profiling of token type preferences (Ge et al., 2023), retention of heavy-hitter tokens (Zhang et al., 2024b;a), and dropping tokens based on the high $L_2$ norms of their key embeddings (Devoto et al., 2024). The latter approach (Devoto et al., 2024) is intriguing as eviction decisions are performed pre-attention. However, this $L_2$ dropout strategy in inclined towards long-context retrieval tasks. It developed based on an empirical observation that smaller norm of key

embedding correlates with higher attention score. For long-context retrieval tasks, high-attention score tokens are the most important tokens since the question's text will overlap with the piece of context that needs to be retrieved. Thus, it is specialized to retain only those tokens with the highest attention, which we find unsuitable for free response reasoning tasks. Existing literature suggests that retaining tokens with a diverse spectrum of attention scores (skewing high) is necessary (Guo et al., 2024; Zhang et al., 2024b; Long et al., 2023).

*Is there a non-attentive KV cache compression strategy that is performant over a wide variety of tasks, including multiple-choice, summarization, long-context retrieval, and free response question-answering?* This work answers this question positively by introducing a novel strategy, LSH-E, that *dynamically* determines token eviction pre-attention via locality-sensitive hashing (LSH) (Goemans & Williamson, 1995; Charikar, 2002). LSH-E evicts a past token from the cache whose key embedding is highly cosine dissimilar to the current query token embedding. The intuition behind this strategy is that high cosine dissimilarity indicates a low dot-product attention score. To efficiently scan for cosine (dis)similar tokens without performing attention, LSH-E leverages the SimHash (Charikar, 2002; Goemans & Williamson, 1995) to instead compare Hamming distances between $c$-length binary hashes of cached key embeddings and the current query embedding. We depict a high-level visualization of this strategy in Figure 1.

LSH-E requires minimal overhead: for a total sequence length of $\ell$ tokens with embedding dimension $d$, LSH-E maintains a constant-size, low-cost binary array in GPU memory of size $c \times k$ bytes, where $c \ll d$ is the hash dimension and $k \ll \ell$. Cached tokens with key embeddings that register low Hamming similarity measurements to decoded query embeddings are gradually replaced.

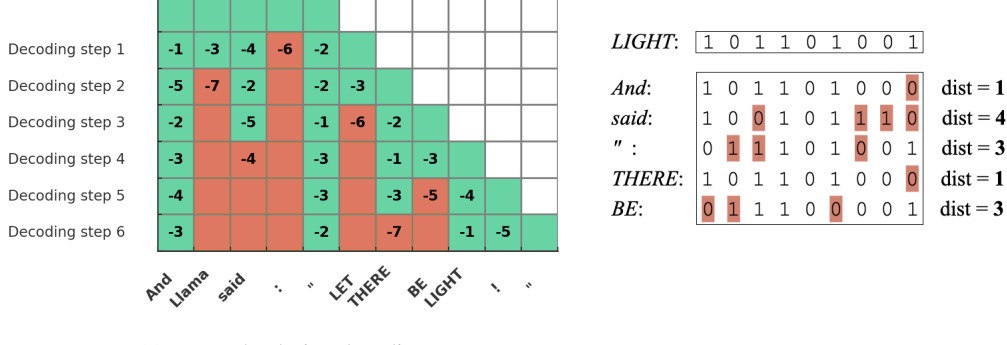

(a) KV cache during decoding  (b) LSH comparison at decoding step 4

Figure 1: **An abstract visualization of LSH-E eviction strategy.** Figure 1a depicts the strategy for several decoding steps. The cache can only maintain 5 tokens due to memory constraints. At each decoding step, LSH-E projects the query embedding of the current token $i$ and all previous key embeddings to *binary hash codes*. LSH-E then measures the negative of Hamming distances between the query code of token $i$ and key codes of all tokens $j$ in the cache. Each step, LSH-E evicts the key/values of the token with the lowest score (marked as red) from the cache. Figure 1b depicts the LSH comparison for decoding step 4, marking the token *"said"* for removal, as its high Hamming indicates low cosine similarity (and thus, low attention).

Our contributions are as follows:

- **Novel Attention-Free Token Eviction** We introduce a novel *attention-free* token eviction strategy, LSH-E, that leverages locality-sensitive hashing (LSH) to quickly locate which token in the cache is the least relevant to the current query. This ranking procedure consists entirely of cheap Hamming distance calculations. The associated binary array for computing these similarities requires minimal memory overhead. For a Llama 3 model, LSH-E can compress the KV cache by 30%-70% with minimal performance drop
- **State-of-the-Art Performance** LSH-E demonstrates high performance on reasoning tasks (GSM8K Cobbe et al. (2021), MedQA Cobbe et al. (2021)), multiple-choice (GSM8K MC, MedQA MC), long-context retrieval (Needle-in-a-Haystack, Common Word (Hsieh et al., 2024)), and long-text summarization (MultiNews, GovReport Bai et al. (2023)). To the best of our knowledge, LSH-E achieves state-of-the-art performance for attention-free eviction, outperforming

the similar attention-free $L_2$ method. Additionally, LSH-E outperforms attention-accumulation-based methods on long text summarization tasks and achieves 1.5x speedup in the prefilling stage and comparable speed in the decoding stage withoug low-level optimiations.

- **Open-Source Implementation** Upon public release of our manuscript, we will release an open-source implementation of LSH-E through a fork of the popular cold-compress library (`https://github.com/AnswerDotAI/cold-compress`).

## 2 PRELIMINARIES

We aim to capture tokens whose query embeddings will form a large sum of dot products (i.e., attention scores) with other key embeddings, but without explicitly calculating attention. We will leverage locality-sensitive hashing (LSH) to quickly determine cosine similarities since the angle is equivalent to the dot product (for unit vectors). In this section, we review technical concepts crucial to attention and locality-sensitive hashing. We assume some base level of similarity with transformers, but we refer the reader to precise formalism (Phuong & Hutter, 2022).

**Scaled Dot-Product Attention** Consider a sequence of $n$ tokens with $e$-dimensional real-valued representations $x_1, x_2, \ldots, x_n$. Let $Q = [q_1 \, q_2 \, \cdots \, q_n] \in \mathbb{R}^{n \times d}$, $K = [k_1 \, k_2 \, \cdots \, k_n] \in \mathbb{R}^{d \times n}$ where $q_i = W_q x_i$, $k_i = W_k x_i$ and $W, K \in \mathbb{R}^{d \times e}$. The query and key projectors $W_q$ and $W_k$ are pre-trained weight matrices. We also define a value matrix $V = [v_1 \, v_2 \, v_2 \, \cdots \, v_n] \in \mathbb{R}^{d_{out} \times n}$ with $v_i = W_v x_i$ with trainable $V \in \mathbb{R}^{d_{out} \times d}$, the scaled dot-product attention mechanism is given as

$$\text{Attention}(Q, K, V) = V \cdot \text{softmax}\Big(\frac{Q^\top K}{\sqrt{d}}\Big). \tag{1}$$

Typically, attention layers contain multiple heads $\{h_i\}_{i=1}^J$ each with distinct query, key, and value projectors $\{W_q^{(h_i)}, W_k^{(h_i)}, W_v^{(h_i)}\}_{i=1}^J$. In a multi-head setup, attention is computed in parallel across all heads, and the outputs are concatenated together and then passed through a linear layer for processing by the next transformer block.

As $Q, K, V$ are updated with each new incoming token, to avoid significant re-computation, the current state of $Q^\top K$, $Q$, and $K$ are maintained in the KV cache. Our goal is to bypass attention computation and caching for select tokens, i.e., sparsify the attention matrix $Q^\top K$, $K$, and $V$.

**Locality-Sensitive Hashing** We will now describe a family of locality-sensitive hashing (LSH) functions able to efficiently approximate nearest neighbors (per cosine similarity) of key/query vectors in high-dimensional $\mathbb{R}^d$ through comparison in a reduced $c$-dimensional space (per Hamming distance) with $c \ll d$. Here, "locality-sensitive" means points that are close together according to a distance function $\text{dist}_d(\cdot, \cdot)$ in the ambient space remain close per another distance function $\text{dist}_c(\cdot, \cdot)$ in the lower-dimensional space with high-probability. For a rigorous treatment of LSH functions, see (Andoni et al., 2018; Charikar, 2002).

Formally for our setup, $dist_d(x, y) \triangleq \cos \theta_{x,y} = \frac{x^\top y}{||x|| \, ||y||}$ and $dist_c(p, q) \triangleq d_H(p, q)$ which denotes the Hamming distance. We will project each vector from $\mathbb{R}^d$ into $\mathbb{Z}_2^c$, the space of $c$-bit binary strings (which is often referred to as a *binary hash code*). To acquire a $c$-bit long hash code from an input vector $x \in \mathbb{R}^d$, we define a random projection matrix $R \in \mathbb{R}^{c \times d}$ whose entries are independently sampled from the standard normal distribution $\mathcal{N}(0, 1)$. We then define

$$h(x) = \text{sgn}(Rx), \tag{2}$$

where $\text{sgn}(\cdot)$ (as an abuse of conventional notation) is the element-wise Heaviside step function:

$$\text{sgn}(x) := \begin{cases} 1, & x \geq 0 \\ 0, & x < 0 \end{cases}.$$

For two unit vectors $x, y \in \mathbb{R}^d$ we have that,

$$\frac{1}{c} \cdot \mathbb{E}[d_H\big(h(x), h(y)\big)] = \frac{\theta_{x,y}}{\pi}, \tag{3}$$

where $\theta_{x,y} = \arccos(\cos(\theta_{x,y}))$. We do not prove equation 3 in this work; see Theorem §3.1 in (Goemans & Williamson, 1995, Theorem 3.1). In particular, if $x$ and $y$ are close in angle, the Hamming distance between $h(x)$ and $h(x)$ is low in expectation. Increasing the hash dimension $c$ reduces variance.

The geometric intuition behind this LSH scheme is the following: each row $R_{:,i}$ of $R$ defines a random hyperplane in $\mathbb{R}^d$. The Heaviside function $\text{sgn}(\cdot)$ indicates whether $x$ is positively or negatively oriented with respect to the hyperplane $R_{:,i}$. Thus, the $c$ hyperplanes divide the $d$ dimensional space into multiple partitions, and the resulting $c$-dimensional hash code is an index into one of the partitions in which $x$ is located. Therefore, vectors with the same or similar hash codes lie in the same or close-by partitions and, therefore, are likely similar in angle. cwecwasdf

## 2.1 RELATED WORKS

**KV Cache Compression**    Many popular compression strategies adopt an *eviction* approach, which removes embeddings from the KV cache. H$_2$O (Zhang et al., 2024b) and Scissorhands (Liu et al., 2024b) calculate token importance by their accumulated attention scores and keep the "heavy hitters" in the cache. FastGen (Ge et al., 2023) performs a profiling pass before the generation stage that assigns to each head, according to the head's attention patterns, a pruning policy which only retains categories of tokens (punctuation, special, etc.) favored by the head. These eviction strategies depend on the computation of attention scores for their policy. An attention-free $L_2$ dropout method (Devoto et al., 2024), which we compare ourselves to in this work, uses the observation that high-attention tokens tend to have low $L_2$ key norms to approximately keep important tokens in cache. Other methods seek to merge KV caches across heads, such as grouped query attention (GQA) (Ainslie et al., 2023; Dubey et al., 2024). KVMerger (Wang et al., 2024) and MiniCache (Liu et al., 2024a), which searches for similarity between tokens in consecutive attention layers and subsequently merges KV cache entries across these layers. While these consolidation approaches prevent memory complexity associated with KV caches from scaling with depth or multi-head attention, the size of any singular cache still tends to scale with sequence length.

**LSH Based Attention**    Similar to our work, Reformer (Kitaev et al., 2020) employs LSH to find similar tokens, but as a way to replace the softmax attention as opposed to token eviction. It creates hash buckets of tokens that form local attention groups and only attends to tokens in the same and neighboring buckets. However, this makes Reformer vulnerable to missing important tokens due to hash collision or boundary issues, and therefore, it must use multiple hash tables to mitigate this issue. In a similar vein, KDEFormer (Zandieh et al., 2023), HyperAttention (Han et al., 2023), and Zandieh et al. (2024a), use LSH to stably approximate and compressing the attention module thus accelerating the computation, but without token eviction. SubGen (Zandieh et al., 2024b) uses LSH to cluster key embeddings and samples representatives from each cluster to reduce the size of the KV Cache and consequently speed up attention, though it must initially view all queries and keys to perform this clustering which could result in VRAM blowup, which our method avoids.

## 3 LSH-E: A LOCALITY-SENSITIVE EVICTION STRATEGY

We now formalize our eviction method reflected in Algorithm 1. We assume that the KV cache has a limited and fixed budget and conceptually divide the KV cache management during LLM inference into two stages: the initial Prompt Encoding Stage and then a Decoding Stage (i.e., generation).

Let $C$ be a constant and fixed cache budget, $\mathcal{K}$ be the key cache, and $\mathcal{V}$ be the V cache in a K-V attention head. We define our eviction policy as a function

$$\mathcal{K}_t, \mathcal{V}_t, \mathcal{H}_t \leftarrow P(q, \mathcal{K}_{t-1}, \mathcal{V}_{t-1}, \mathcal{H}_{t-1}) \tag{4}$$

where $\mathcal{H}_t \in \{0,1\}^{b \times C}$ is a hash table that contains hash codes of keys in $\mathcal{K}$. We then define a function $F_{score}$ to assign a score for each key inside the K cache. $F_{score}$ outputs an array which contains the negative of hamming distances $d_H$ between the hash code of a query vector $q$ and columns of $\mathcal{H}$, which are hash codes of all non-evicted keys.

$$F_{score}(q, \mathcal{K}) = -d_H(h(q), \mathcal{H}) \tag{5}$$

The eviction index $e_t$ at any step $t$ is selected as the index with the lowest score:

$$e_t \leftarrow \arg\min F_{score}(q_{t-1}, \mathcal{H}_{t-1}) \tag{6}$$

which points to the key that is most distant from the query vector at time step $t$. Entries at index $e_t$ from the $\mathcal{K}$ and $\mathcal{V}$ are evicted and $\mathcal{H}$ is updated (step 3-6 of Algorithm 1).

---

**Algorithm 1** LSH-E (timestep $t$)

---

**Require:** query $q$, key $k$, value $v$, key cache $\mathcal{K}$, value cache $\mathcal{V}$, hash table $\mathcal{H}$
1: $e_t \leftarrow \arg\min F_{score}(q_t, \mathcal{H}_{t-1})$ ▷ Determine eviction index $e_t$
2: **del** $\mathcal{K}_{t-1}^{e_t}, \mathcal{V}_{t-1}^{e_t}, \mathcal{H}_{t-1}^{e_t}$ ▷ Remove entries at index $e_t$ from KV cache and hash table
3: $\mathcal{K}_t \leftarrow \mathcal{K}_{t-1} \cup k_t$ ▷ Update key cache
4: $\mathcal{V}_t \leftarrow \mathcal{V}_{t-1} \cup v_t$ ▷ Update value cache
5: $\mathcal{H}_t \leftarrow \mathcal{H}_{t-1} \cup h(k_t)$ ▷ Add hash of $k_t$ to the hash table
6: $A \leftarrow \text{Attention}(q, \mathcal{K}_T, \mathcal{V}_T)$ ▷ Calculate attention

---

**Prompt Encoding Stage** During the prompt encoding stage, the model processes the prompt, $x_{prompt} = [x_1, ..., x_N] \in \mathbb{R}^{N \times d}$. The KV cache and the hash table are first filled to full by the first $C$ tokens. $\mathcal{K}_0 = \{k_1, ..., k_C\}, \mathcal{V}_0 = \{v_1, ..., v_C\}, \mathcal{H}_0 = h(\mathcal{K}_0) = \bigcup_{i \in [1,C]} h(k_i)$. We then set $t \leftarrow C + 1$, and begin Algorithm 1.

**Decoding Stage** Let $x_{decoding} = [z_1, ... z_T] \in \mathbb{R}^{T \times d}$ be the generated tokens during auto-regressive decoding. In the decoding stage, we continue Algorithm 1 by setting $t < -N + 1$. The generation completes at time step $N + T$.

**Complexity** Our strategy assumes a fixed memory budget, and therefore, uses constant memory. The computation overhead per time step is also constant, because $F_score$ is calculated for a constant $C$ number of key vectors in the cache. The extra memory overhead that LSH-E introduces to each attention head is the hash table $\mathcal{H}$, which only uses $C * b$ bits of space and is independent of the sequence length. The hash table is stored on GPU memory and does not introduce any latency bottlenecks associated with CPU-to-GPU streaming (Strati et al., 2024).

## 4 EXPERIMENTS

**Tasks** We evaluated our LSH eviction strategy across various tasks to demonstrate its effectiveness in reducing the memory cost of the KV cache while preserving the language quality of the generated text. Our experiments are split into four main categories: free response question answering, multiple choice, long-context retrieval and long-context summarization. Our long context retrieval tasks include the multi-key needle-in-a-haystack task and the common words task from (Hsieh et al., 2024). Question answering tasks include GSM8K (Cobbe et al., 2021) and MedQA (Jin et al., 2021). Summarizaiton tasks include GovReport and MultiNews from Bai et al. (2023).

**Metrics** The question-answering tasks were evaluated using BERTScore (which includes precision, recall, and F1 scores), ROUGE (ROUGE-1, ROUGE-2 and ROUGE-L and ROUGE-Lsum), and GPT4-Judge. GPT-4 was prompted to look at both the model prediction and the ground truth answer, then provide a score from 1 - 5 on the coherence, faithfulness, and helpfulness of the answer in addition to similarity between the prediction and ground truth (we named this metric GPT4-Rouge). In this section, we report the average of these four scores. For details on individual scores, please see Appendix B. For the system prompts given to GPT-4, refer to Appendix G.2. For multiple-choice tasks, we use accuracy as our metric. The metric used to evaluate long context retrieval tasks is the string matching score from Hsieh et al. (2024), whose definition is in Appendix G.1. For summarization tasks, we use Rouge as the metric as per direction from Bai et al. (2023).

**Configuration and Setup** We conducted most experiments using Meta's Llama3 8B-Instruct model (Dubey et al., 2024) with the exception of long text summarization tasks which were tested using the Llama3.1 8B-Instruct model. Our method is agnostic to grouped-query attention, so we used the default group size of 4. The maximum sequence length was set to the sum of the maximum

prompt length and the maximum number of allowed generated tokens needed for each task. We conducted experiments using cache budgets of 10%, 30%, 50%, 70%, and 90% of the full KV cache. Based on insights from (Xiao et al., 2023; Child et al., 2019; Beltagy et al., 2020), we also keep the most recent 10 tokens and the first 4 tokens of the prompt always in the KV cache. The summarization tasks were performed on Nvidia H100 80GB graphics cards due to their long contexts. All other experiments were conducted on the Google Cloud Platform G2 instances with Nvidia L4 24GB graphics cards.

**Baseline Methods** We chose the $L_2$ norm-based eviction method (Devoto et al., 2024) as our main baseline for comparison because it is also an attention-free KV cache eviction method. We also included two attention-accumulation-based methods: H₂O Zhang et al. (2024b) and Scissorhands Liu et al. (2024b), as well as a hybrid method: Fastgen Ge et al. (2023).

## 4.1 FREE RESPONSE QUESTION ANSWERING

We tested our strategy against tasks that require generating accurate answers using multi-step reasoning. Specifically, we used the GSM8K and MedQA datasets to assess language quality for each strategy, given a constrained KV cache budget. Both tasks are used to test the potential side effects of compression on the LLM's reasoning ability.

**GSM8K** GSM8K consists of grade-school-level math problems that typically require multiple reasoning steps. As shown in Figure 2, our LSH eviction strategy consistently outperforms the $L_2$ norm-based method across various cache sizes. Notably, even when the KV cache budget is set to 50% of the full capacity, the LSH eviction strategy maintains a high answer quality, with minimal degradation in BERTScore F1, ROUGE-L, and GPT4-Judge scores. Additionally, LSH-E performs on par with H2O and Scissorhands without accumulating attention scores.

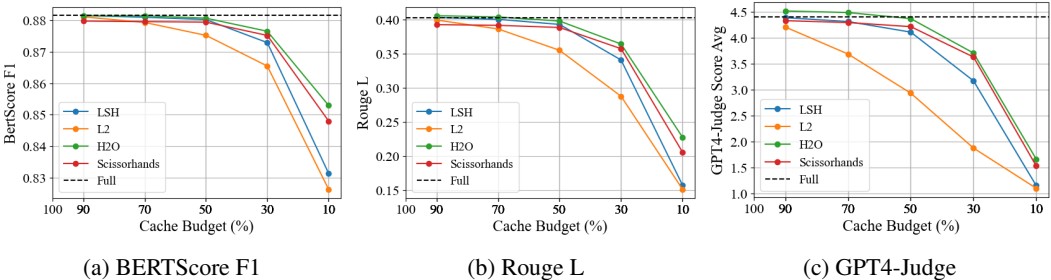

|                  |                  |                  |
|:----------------:|:----------------:|:----------------:|
| (a) BERTScore F1 | (b) Rouge L      | (c) GPT4-Judge   |

Figure 2: **GSM8K Question Answering Performance.** We measure BERTScore F1, Rouge-L, and GPT4-Judge for different cache budgets on a grade school math task. LSH-E outperforms $L_2$ for all three metrics for every budget, with sharp differences for the 50% and 30% compression. LSH-E performs similarly to H₂O and Scissorhands except at 10% cache budget.

**MedQA** MedQA is a free response multiple choice question answering dataset collected from professional medical board exams. We randomly sampled 100 questions from this dataset. Each question has 5 choices and only one correct answer, along with ground truth explanations and reasoning steps. Figure 3 illustrates that LSH-E performs better than all baseline methods for all cache budgets tested. For both datasets, LSH-E produced more coherent and helpful answers across all cache budgets than the baselines per Table 8.

For detailed experiment results of both question anwering tasks, and for comparison with Fastgen at various attention recovery ratios, please refer to Appendix B.

## 4.2 MULTIPLE CHOICE QUESTION ANSWERING

We evaluated our method on multiple-choice versions of GSM8K and MedQA. Multiple choice is a more difficult test of a model's reasoning capability under the constraint of cache compression, as it takes away the ability to use intermediate results in the generated text. The model has to keep useful tokens during prompt compression in order to pick the correct answer choice.

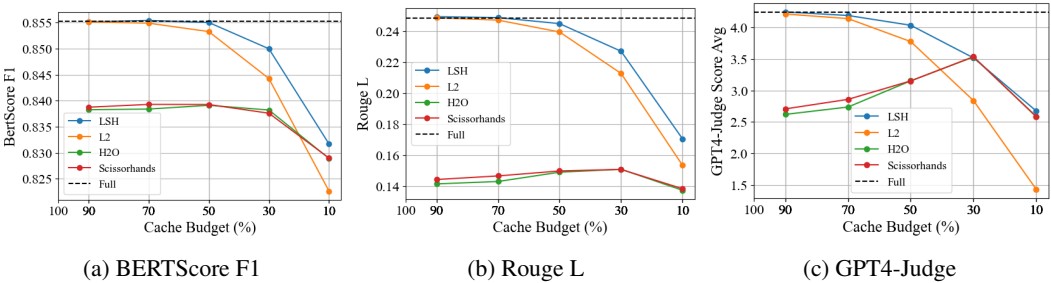

(a) BERTScore F1     (b) Rouge L     (c) GPT4-Judge

Figure 3: **MedQA Question Answering Performance.** We measure BertScore F1, Rouge-L, and GPT4-Judge for different cache budgets on a medical exam task. LSH outperforms $L_2$ for all three metrics for every budget, with a significantly higher performance for the 30% and 10% budgets.

**GSM8K Multiple Choice** For the GSM8K multiple choice experiments, LSH significantly out-performs $L_2$ for cache budgets of 30% and 50%. As shown in Figure 4a, the $L_2$ method's accuracy drops significantly at smaller cache sizes, while the performance of LSH-E does not significantly drop until the cache budget is set at 10%.

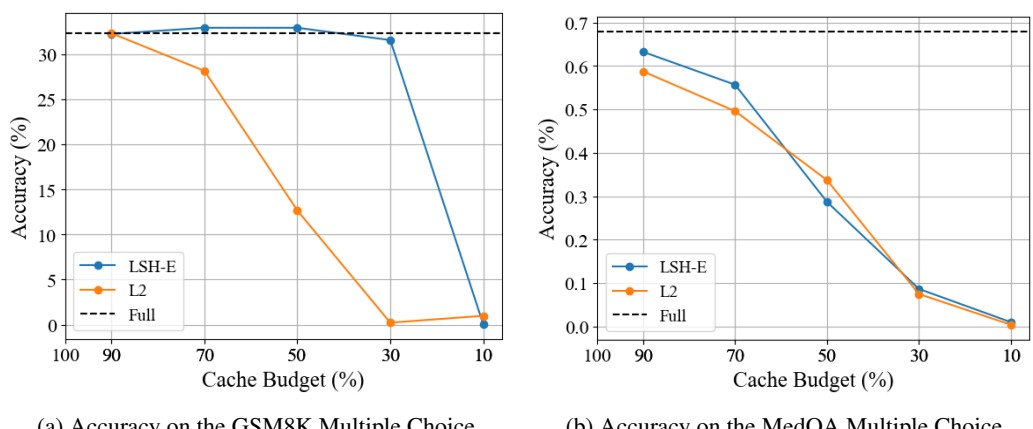

(a) Accuracy on the GSM8K Multiple Choice   (b) Accuracy on the MedQA Multiple Choice

Figure 4: **Multiple Choice Tasks Performance.** On GSM8K, LSH-E outperforms the baseline full cache on GSM8K at 70% and 50% cache budgets and significantly outperforms $L_2$ at 70%, 50%, and 30%. LSH-E performs on par with $L_2$ overall on MedQA with higher performance at 90% (near uncompressed performance) and 70% budget and slightly lower performance at 50% budget.

**MedQA Multiple Choice** Per Figure 4b, the MedQA multiple choice experiment, LSH offers better performance than $L_2$ eviction for all tested cache budgets except for 50%. Performance between both methods is highly similar at lower budgets.

### 4.3 LONG-CONTEXT RETRIEVAL

To evaluate LSH-E's ability to retain and retrieve important pieces of information from long con-texts, we used the Needle-in-a-Haystack and Common Words tasks from Hsieh et al. (2024) with 4K context length. These tests benchmark the ability of a compression strategy to retain important tokens inside the KV cache within a large, complex stream of context.

**Needle-in-a-Haystack** In the Needle-in-a-Haystack task, the model must extract specific informa-tion buried within a large body of text. As illustrated in Figure 5b, LSH-E slightly outperforms $L_2$ at every cache budget except for 90%, and both methods see a sharp drop in the ability to recall the "needle" (a small, targeted piece of context) after the cache budget drops to 50% and lower. LSH-E outperforms $L_2$ for these smaller cache sizes.

**Common Words** In the Common Words task, the model must identify the most frequent words from a long list. Figure 5a demonstrates that LSH-E performs on par with $L_2$ eviction in general and slightly better at 30%, 50%, and 90% cache budget. Both methods outperform the full cache model at 90% cache size, indicating that some cache compression can actually increase performance. Neither method experienced a significant drop in performance until the cache budget was reduced to 30%.

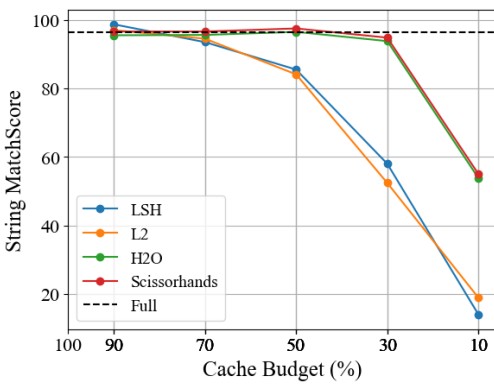 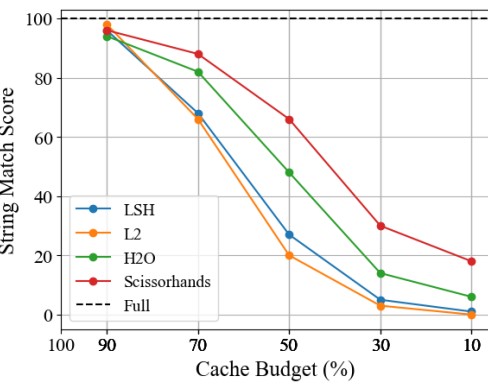

(a) String Match Score on Common Words    (b) String Match Score on Needle-in-a-Haystack

Figure 5: **Long-Context Tasks.** We measure string-matching scores for two long-context retrieval tasks. LSH-E performs on par with $L_2$ on the Common Words task with slightly higher performance at a 30% cache budget and slightly lower performance at a 10% budget. For the Needle-in-a-Haystack task, LSH-E performs on par with $L_2$ with slightly higher performance at a 50% cache budget.

### 4.4 LONG TEXT SUMMARIZATION

To evaluate LSH-E's ability to handle exceptionally long context lengths, we incorporated the Multi-News and GovReport summarizations tasks from LongBench Bai et al. (2023). We tested both tasks using the Llama3.1-8B-Instruct model and used context size of 16K tokens.

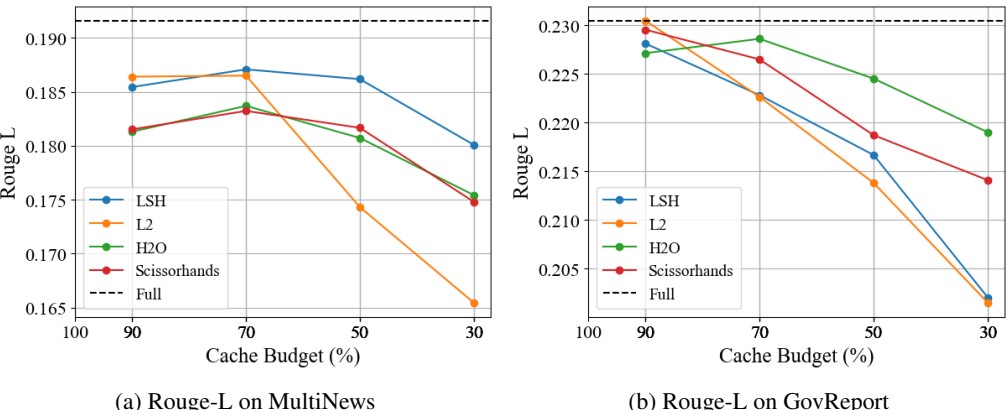

(a) Rouge-L on MultiNews    (b) Rouge-L on GovReport

Figure 6: **LongBench Summarization Tasks** We measure Rouge-L for two long text summarization tasks. LSH-E outperforms all baseline methods on MultiNews at 30 - 70% cache budget. LSH-E performs better than $L_2$ on GovReport at 50% cache budget similarly at 30% and 70%.

**MultiNews** The MultiNews dataset contains clusters of 2-10 news articles discussing the same event or topic. The model is asked to provide a one-page summary of the articles. LSH-E outperforms all baselines in the MultiNews summarization task at 30-70% cache budget. At 90% cache budget, LSH-E still outperforms $H_2O$ and Scissorhands while being slighly lower thant $L_2$.

**GovReport** The GovReport dataset contains reports spanning a wide variety of national policy issues from the U.S. Government. The model is asked to produce a one-page summary of the reports. LSH-E performs on par with and sometimes slightly better than $L_2$ at 30-70% cache budget, while not as well as $H_2O$ or Scissorhands.

## 4.5 THROUGHPUT

To evaluate the speed of LSH-E and baseline methods, we measured the decoding and prefilling speed during the MultiNews evaluation. Because the length of answers generated by each eviction strategy generates can be different, we report decoding and prefilling speed in tokens per second instead of elapsed time.

Table 1: **Throughput on LongBench MultiNews Summarization Task** LSH-E method is as fast as $L_2$ and faster than other baselines at both prefilling and decoding, even without low-level optimizations (i.e., expressing our hash tables in true binary bits). At the prefill stage, LSH-E is 1.5x as fast as $H_2O$ and Scissorhands and 17x as fast compared to FastGen.

| Cache Budget (%) / Fastgen Attn Recovery Frac (%) | Strategy | Rouge L | Decode Toks Per Sec | Prefill Tokes Per Sec |
|---|---|---|---|---|
| 30 | LSH-E | 0.180 | 22.880 | 20293.524 |
| | $L_2$ | 0.165 | 23.981 | 20628.160 |
| | $H_2O$ | 0.175 | 21.555 | 13025.776 |
| | Scissorhands | 0.175 | 21.448 | 13004.254 |
| 50 | LSH-E | 0.186 | 22.846 | 20459.961 |
| | $L_2$ | 0.174 | 16.013 | 15851.952 |
| | $H_2O$ | 0.181 | 21.973 | 13969.985 |
| | Scissorhands | 0.182 | 20.978 | 13549.967 |
| 70 | LSH-E | 0.187 | 22.914 | 21002.334 |
| | $L_2$ | 0.187 | 24.305 | 21303.763 |
| | $H_2O$ | 0.184 | 21.793 | 14050.521 |
| | Scissorhands | 0.183 | 21.705 | 13954.693 |
| 90 | LSH-E | 0.185 | 22.873 | 21229.230 |
| | $L_2$ | 0.186 | 24.010 | 21305.693 |
| | $H_2O$ | 0.181 | 21.665 | 14007.697 |
| | Scissorhands | 0.182 | 21.411 | 14025.440 |
| 100 | Full | 0.192 | 16.071 | 16573.492 |
| 70 | Fastgen | 0.129 | 12.752 | 1171.069 |
| 75 | | 0.174 | 12.291 | 1157.987 |
| 80 | | 0.184 | 11.850 | 1142.679 |
| 85 | | 0.183 | 11.658 | 1164.689 |

## 4.6 MEMORY USAGE

Table 2 compares the memory usage of the KV cache and relevant data structures of $L_2$ and LSH-E on the GSM8K and MedQA question answering experiments. LSH-E maintains $\mathcal{H}$, a binary hash matrix of the attention keys in memory and, therefore, has slightly higher memory usage than $L_2$ eviction. Our implementation uses 8 bits for binary values instead of 1 bit. Using 1-bit binary numbers would reduce the memory overhead of LSH-E by a factor of 8 and narrow the difference in memory usage between LSH-E and $L_2$.

## 4.7 ABLATION ON LSH DIMENSION

To determine the effect of the LSH compression dimension, we conducted an ablation study using the GSM8K free response dataset. Fixing the cache budget to 50%, we tested LSH dimensions of 4,

Table 2: **GSM8K and MedQA Question Answering KV Cache Memory Usage.** LSH-E maintains a binary hash matrix of attention keys in memory and, therefore, has slightly higher memory usage than $L_2$. Our implementation uses 8-bits for binary values instead of 1-bit. Using 1-bit binary numbers will reduce the memory overhead of LSH-E by a factor of 8 and decrease the difference in memory usage between LSH-E and $L_2$.

| Cache Budget (%) | Strategy | GSM8K | | MedQA | |
|---|---|---|---|---|---|
| | | Compression Ratio | Cache Memory (GB) | Compression Ratio | Cache Memory (GB) |
| 10 | $L_2$ | 0.8355 | 0.7603 | 0.9289 | 2.5342 |
| | LSH-E | 0.8380 | 0.8120 | 0.8812 | 2.6338 |
| 30 | $L_2$ | 0.6234 | 1.7740 | 0.6957 | 7.3492 |
| | LSH-E | 0.6018 | 1.8531 | 0.6360 | 7.5786 |
| 50 | $L_2$ | 0.3968 | 2.7876 | 0.4175 | 12.1641 |
| | LSH-E | 0.3716 | 2.8941 | 0.3901 | 12.5235 |
| 70 | $L_2$ | 0.1967 | 3.8013 | 0.1803 | 17.2325 |
| | LSH-E | 0.1857 | 3.9351 | 0.1740 | 17.7285 |
| 90 | $L_2$ | 0.0859 | 4.8150 | 0.0498 | 22.0474 |
| | LSH-E | 0.0823 | 4.9761 | 0.0483 | 22.6734 |
| 100 | Full | 0.0000 | 12.6934 | 0.0000 | 51.1181 |

8, 16, 32 and 64 bits. The choice of LSH dimension does not significantly impact performance. In fact, 8 bits performed the best, but not noticeably better than higher dimensions. This demonstrates that LSH-E does not require a high hashing dimension and can be executed with minimal storage overhead. When using 8 bits, the storage overhead is 1 byte $\times$ cache size. For example, in a Llama3 70B-Instruct deployment with 80 layers, 8 KV-heads, sequence length of 8192, batch size of 8 and 50% cache budget, LSH dimension of 8-bits, we have that 16-bits and 32-bits only use an extra 20MB, 40MB, and 80MB respectively, which are significantly smaller than the KV cache size of 640GB. Detailed results can be found in Table 9 of Appendix C.

## 5    DISCUSSION & CONCLUSION

In this paper, we introduce LSH-E, a novel attention-free eviction strategy for KV cache compression in transformer-based LLMs. By leveraging locality-sensitive hashing (LSH) to approximate cosine similarity, LSH-E dynamically determines which tokens to evict from the cache without performing costly attention calculations. Our experiments demonstrate that LSH-E can achieve 30-70% compression of the KV cache while maintaining strong performance across various tasks, including free-response Q&A, multiple-choice Q&A, and long-context retrieval.

The key advantage of LSH-E lies in its ability to efficiently compress the KV cache pre-attention, enabling significant memory savings and faster inference times. Compared to traditional strategies like $L_2$ norm-based eviction (Devoto et al., 2024), LSH-E excels particularly in reasoning and multiple-choice tasks, where maintaining a diverse set of tokens in the cache is crucial for generating accurate and coherent responses.

There are several potential areas for future work. Investigating hybrid approaches that combine LSH-based eviction with attention-based mechanisms such as (Zhang et al., 2024b; Ge et al., 2023) could offer a middle ground between computational efficiency and retention of high-importance tokens. Further, reducing the overhead associated with maintaining binary hash codes (e.g., by optimizing bit precision) could further enhance the applicability of LSH-E to memory-constrained environments.

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

APPENDIX

## A  FURTHER RELATED WORKS

**Memory Efficient Transformers**  Multi-Query Attention (Shazeer, 2019) and Grouped Query Attention (Ainslie et al., 2023) reduce the number of key-value matrices by sharing them across multiple query heads to save KV cache memory usage. However, they require re-training or up-training the LLM. Cache quantization methods (Hooper et al., 2024; Sheng et al., 2023) reduce the KV cache size by compressing the hidden dimension instead of along the sequence dimension but can result in information loss. Linear Transformer (Katharopoulos et al., 2020) reduces memory usage by replacing the softmax attention with linear kernels and, therefore, achieves constant memory requirement

## B  QUESTION ANSWERING GRANULAR EXPERIMENT RESULTS

Table 3: GSM8K and MedQA Question Answering BERTScore

| Cache Budget / Fastgen Attn Recovery Frac (%) | Strategy | GSM8K | | | Medqa | | |
| --- | --- | --- | --- | --- | --- | --- | --- |
| | | Precision | Recall | F1 | Precision | Recall | F1 |
| 10 | LSH-E | 0.859 | 0.806 | 0.831 | 0.857 | 0.808 | 0.832 |
| | $L_2$ | 0.858 | 0.798 | 0.826 | 0.833 | 0.813 | 0.823 |
| | $H_2O$ | 0.877 | 0.830 | 0.853 | 0.866 | 0.795 | 0.829 |
| | Scissorhands | 0.873 | 0.825 | 0.848 | 0.867 | 0.795 | 0.829 |
| 30 | LSH-E | 0.893 | 0.854 | 0.873 | 0.867 | 0.834 | 0.850 |
| | $L_2$ | 0.885 | 0.847 | 0.865 | 0.855 | 0.834 | 0.844 |
| | $H_2O$ | 0.893 | 0.860 | 0.877 | 0.878 | 0.802 | 0.838 |
| | Scissorhands | 0.893 | 0.858 | 0.875 | 0.877 | 0.802 | 0.838 |
| 50 | LSH-E | 0.897 | 0.865 | 0.880 | 0.869 | 0.842 | 0.855 |
| | $L_2$ | 0.891 | 0.861 | 0.875 | 0.866 | 0.841 | 0.853 |
| | $H_2O$ | 0.896 | 0.866 | 0.881 | 0.879 | 0.803 | 0.839 |
| | Scissorhands | 0.896 | 0.864 | 0.879 | 0.878 | 0.804 | 0.839 |
| 70 | LSH-E | 0.896 | 0.866 | 0.881 | 0.869 | 0.843 | 0.855 |
| | $L_2$ | 0.894 | 0.865 | 0.879 | 0.868 | 0.842 | 0.855 |
| | $H_2O$ | 0.897 | 0.867 | 0.881 | 0.879 | 0.801 | 0.838 |
| | Scissorhands | 0.896 | 0.864 | 0.880 | 0.879 | 0.803 | 0.839 |
| 90 | LSH-E | 0.897 | 0.867 | 0.881 | 0.868 | 0.843 | 0.855 |
| | $L_2$ | 0.896 | 0.866 | 0.881 | 0.868 | 0.843 | 0.855 |
| | $H_2O$ | 0.897 | 0.867 | 0.881 | 0.879 | 0.801 | 0.838 |
| | Scissorhands | 0.896 | 0.864 | 0.880 | 0.880 | 0.802 | 0.839 |
| 50 | Fastgen | 0.811 | 0.770 | 0.789 | 0.816 | 0.763 | 0.788 |
| 60 | | 0.827 | 0.778 | 0.801 | 0.806 | 0.766 | 0.785 |
| 70 | | 0.837 | 0.788 | 0.811 | 0.811 | 0.766 | 0.787 |
| 80 | | 0.874 | 0.840 | 0.857 | 0.866 | 0.793 | 0.828 |
| 90 | | 0.896 | 0.864 | 0.879 | 0.876 | 0.800 | 0.836 |
| 100 | Full | 0.897 | 0.867 | 0.882 | 0.868 | 0.843 | 0.855 |

## C  RESULTS OF ABLATION ON LSH DIMENSION

Please see Table 9

Table 4: GSM8K Question Answering Rouge

| Cache Budget / Fastgen Attn Recovery Fra(%) | Strategy | Rouge 1 | Rouge 2 | Rouge L | Rouge Lsum |
|---|---|---|---|---|---|
| 10 | LSH-E | 0.206 | 0.051 | 0.157 | 0.186 |
| | $L_2$ | 0.196 | 0.050 | 0.151 | 0.179 |
| | $H_2O$ | 0.300 | 0.090 | 0.227 | 0.263 |
| | Scissorhands | 0.271 | 0.074 | 0.205 | 0.238 |
| 30 | LSH-E | 0.446 | 0.187 | 0.341 | 0.383 |
| | $L_2$ | 0.392 | 0.149 | 0.288 | 0.337 |
| | $H_2O$ | 0.481 | 0.208 | 0.364 | 0.410 |
| | Scissorhands | 0.471 | 0.203 | 0.357 | 0.403 |
| 50 | LSH-E | 0.511 | 0.234 | 0.393 | 0.438 |
| | $L_2$ | 0.476 | 0.205 | 0.355 | 0.409 |
| | $H_2O$ | 0.517 | 0.238 | 0.398 | 0.442 |
| | Scissorhands | 0.509 | 0.232 | 0.389 | 0.433 |
| 70 | LSH-E | 0.521 | 0.240 | 0.401 | 0.446 |
| | $L_2$ | 0.509 | 0.230 | 0.386 | 0.435 |
| | $H_2O$ | 0.523 | 0.243 | 0.404 | 0.446 |
| | Scissorhands | 0.510 | 0.233 | 0.392 | 0.435 |
| 90 | LSH-E | 0.525 | 0.243 | 0.403 | 0.449 |
| | $L_2$ | 0.522 | 0.241 | 0.400 | 0.446 |
| | $H_2O$ | 0.523 | 0.243 | 0.406 | 0.446 |
| | Scissorhands | 0.512 | 0.235 | 0.393 | 0.436 |
| 50 | Fastgen | 0.112 | 0.017 | 0.095 | 0.106 |
| 60 | | 0.133 | 0.024 | 0.113 | 0.126 |
| 70 | | 0.171 | 0.036 | 0.139 | 0.160 |
| 80 | | 0.356 | 0.128 | 0.264 | 0.305 |
| 90 | | 0.509 | 0.231 | 0.391 | 0.434 |
| 100 | Full | 0.526 | 0.244 | 0.403 | 0.449 |

Table 5: GSM8K Question Answering GPT4-Judge

| Cache Budget / Fastgen Attn Recovery Frac (%) | Strategy | Similarity to GT | Coherence | Faithfulness | Helpfulness |
|---|---|---|---|---|---|
| 10 | LSH-E | 1.018 | 1.387 | 1.147 | 1.083 |
| | $L_2$ | 1.005 | 1.293 | 1.098 | 1.033 |
| | $H_2O$ | 1.172 | 2.304 | 1.566 | 1.630 |
| | Scissorhands | 1.138 | 2.132 | 1.424 | 1.452 |
| 30 | LSH-E | 2.520 | 3.767 | 3.216 | 3.190 |
| | $L_2$ | 1.356 | 2.428 | 1.895 | 1.841 |
| | $H_2O$ | 3.014 | 4.252 | 3.706 | 3.860 |
| | Scissorhands | 2.906 | 4.184 | 3.636 | 3.798 |
| 50 | LSH-E | 3.457 | 4.530 | 4.212 | 4.241 |
| | $L_2$ | 2.190 | 3.494 | 3.035 | 3.027 |
| | $H_2O$ | 3.798 | 4.712 | 4.434 | 4.534 |
| | Scissorhands | 3.582 | 4.604 | 4.276 | 4.400 |
| 70 | LSH-E | 3.734 | 4.671 | 4.404 | 4.444 |
| | $L_2$ | 2.934 | 4.184 | 3.817 | 3.820 |
| | $H_2O$ | 3.940 | 4.774 | 4.576 | 4.656 |
| | Scissorhands | 3.712 | 4.668 | 4.334 | 4.462 |
| 90 | LSH-E | 3.569 | 4.578 | 4.324 | 4.361 |
| | $L_2$ | 3.837 | 4.722 | 4.468 | 4.525 |
| | $H_2O$ | 3.970 | 4.814 | 4.596 | 4.688 |
| | Scissorhands | 3.750 | 4.676 | 4.392 | 4.504 |
| 50 | | 1.000 | 1.074 | 1.040 | 1.028 |
| 60 | | 1.000 | 1.054 | 1.022 | 1.010 |
| 70 | Fastgen | 1.008 | 1.116 | 1.048 | 1.014 |
| 80 | | 1.472 | 2.602 | 2.118 | 2.234 |
| 90 | | 3.838 | 4.714 | 4.448 | 4.554 |
| 100 | Full | 3.845 | 4.716 | 4.499 | 4.545 |

Table 6: MedQA Question Answering BERTScore

| Cache Budget / Fastgen Attn Recovery Fra(%) | Strategy | Precision | Recall | F1 |
|---|---|---|---|---|
| 10 | LSH-E | 0.857 | 0.808 | 0.832 |
| | $L_2$ | 0.833 | 0.813 | 0.823 |
| | $H_2O$ | 0.866 | 0.795 | 0.829 |
| | Scissorhands | 0.867 | 0.795 | 0.829 |
| 30 | LSH-E | 0.867 | 0.834 | 0.850 |
| | $L_2$ | 0.855 | 0.834 | 0.844 |
| | $H_2O$ | 0.878 | 0.802 | 0.838 |
| | Scissorhands | 0.877 | 0.802 | 0.838 |
| 50 | LSH-E | 0.869 | 0.842 | 0.855 |
| | $L_2$ | 0.866 | 0.841 | 0.853 |
| | $H_2O$ | 0.879 | 0.803 | 0.839 |
| | Scissorhands | 0.878 | 0.804 | 0.839 |
| 70 | LSH-E | 0.869 | 0.843 | 0.855 |
| | $L_2$ | 0.868 | 0.842 | 0.855 |
| | $H_2O$ | 0.879 | 0.801 | 0.838 |
| | Scissorhands | 0.879 | 0.803 | 0.839 |
| 90 | LSH-E | 0.868 | 0.843 | 0.855 |
| | $L_2$ | 0.868 | 0.843 | 0.855 |
| | $H_2O$ | 0.879 | 0.801 | 0.838 |
| | Scissorhands | 0.880 | 0.802 | 0.839 |
| 50 | | 0.816 | 0.763 | 0.788 |
| 60 | | 0.806 | 0.766 | 0.785 |
| 70 | Fastgen | 0.811 | 0.766 | 0.787 |
| 80 | | 0.866 | 0.793 | 0.828 |
| 90 | | 0.876 | 0.800 | 0.836 |
| 100 | Full | 0.868 | 0.843 | 0.855 |

Table 7: MedQA Question Answering Rouge

| Cache Budget (%) | Strategy | Rouge 1 | Rouge 2 | Rouge L | Rouge Lsum |
|---|---|---|---|---|---|
| 10 | LSH-E | 0.346 | 0.110 | 0.171 | 0.324 |
| | $L_2$ | 0.304 | 0.072 | 0.154 | 0.289 |
| | $H_2O$ | 0.236 | 0.092 | 0.138 | 0.220 |
| | Scissorhands | 0.237 | 0.091 | 0.139 | 0.221 |
| 30 | LSH-E | 0.449 | 0.170 | 0.227 | 0.426 |
| | $L_2$ | 0.429 | 0.146 | 0.213 | 0.407 |
| | $H_2O$ | 0.255 | 0.118 | 0.151 | 0.239 |
| | Scissorhands | 0.252 | 0.116 | 0.151 | 0.236 |
| 50 | LSH-E | 0.481 | 0.194 | 0.245 | 0.455 |
| | $L_2$ | 0.474 | 0.184 | 0.240 | 0.449 |
| | $H_2O$ | 0.243 | 0.107 | 0.149 | 0.229 |
| | Scissorhands | 0.244 | 0.110 | 0.150 | 0.230 |
| 70 | LSH-E | 0.487 | 0.197 | 0.249 | 0.461 |
| | $L_2$ | 0.484 | 0.194 | 0.247 | 0.458 |
| | $H_2O$ | 0.229 | 0.097 | 0.143 | 0.216 |
| | Scissorhands | 0.234 | 0.103 | 0.147 | 0.219 |
| 90 | LSH-E | 0.487 | 0.197 | 0.249 | 0.461 |
| | $L_2$ | 0.487 | 0.197 | 0.249 | 0.461 |
| | $H_2O$ | 0.223 | 0.095 | 0.142 | 0.211 |
| | Scissorhands | 0.228 | 0.099 | 0.145 | 0.214 |
| 50 | | 0.068 | 0.013 | 0.052 | 0.066 |
| 60 | | 0.079 | 0.014 | 0.061 | 0.077 |
| 70 | Fastgen | 0.103 | 0.020 | 0.074 | 0.099 |
| 80 | | 0.208 | 0.069 | 0.126 | 0.192 |
| 90 | | 0.220 | 0.092 | 0.140 | 0.207 |
| 100 | Full | 0.486 | 0.198 | 0.248 | 0.460 |

Table 8: MedQA Question Answering GPT4-Judge

| Cache Budget / Fastgen Attn Recovery Frac (%) | Strategy | Similarity to GT | Coherence | Faithfulness | Helpfulness |
|---|---|---|---|---|---|
| 10 | LSH-E | 1.970 | 3.517 | 2.665 | 2.547 |
| | $L_2$ | 1.103 | 1.695 | 1.639 | 1.283 |
| | $H_2O$ | 2.138 | 3.206 | 2.594 | 2.416 |
| | Scissorhands | 2.144 | 3.202 | 2.580 | 2.402 |
| 30 | LSH-E | 2.511 | 4.415 | 3.533 | 3.613 |
| | $L_2$ | 1.939 | 3.633 | 2.942 | 2.843 |
| | $H_2O$ | 3.428 | 3.818 | 3.608 | 3.276 |
| | Scissorhands | 3.406 | 3.850 | 3.602 | 3.286 |
| 50 | LSH-E | 3.022 | 4.730 | 4.139 | 4.254 |
| | $L_2$ | 2.850 | 4.511 | 3.797 | 3.950 |
| | $H_2O$ | 2.938 | 3.632 | 3.280 | 2.762 |
| | Scissorhands | 2.918 | 3.634 | 3.308 | 2.748 |
| 70 | LSH-E | 3.232 | 4.809 | 4.292 | 4.434 |
| | $L_2$ | 3.194 | 4.755 | 4.235 | 4.385 |
| | $H_2O$ | 2.414 | 3.396 | 2.958 | 2.178 |
| | Scissorhands | 2.554 | 3.454 | 3.098 | 2.328 |
| 90 | LSH-E | 3.291 | 4.839 | 4.355 | 4.507 |
| | $L_2$ | 3.265 | 4.818 | 4.318 | 4.458 |
| | $H_2O$ | 2.400 | 3.232 | 2.830 | 2.016 |
| | Scissorhands | 2.404 | 3.346 | 2.980 | 2.098 |
| 50 | | 1.002 | 1.004 | 1.006 | 1.000 |
| 60 | | 1.005 | 1.004 | 1.005 | 1.000 |
| 70 | Fastgen | 1.008 | 1.014 | 1.014 | 1.008 |
| 80 | | 1.620 | 2.783 | 2.270 | 1.512 |
| 90 | | 2.356 | 3.242 | 2.748 | 1.870 |
| 100 | Full | 3.337 | 4.817 | 4.342 | 4.500 |

Table 9: **LSH Hash Dimension Ablation.** We assesses GSM8K Question Answering performance for different LSH dimensions. The cache budget is fixed at 50%. LSH dimension does not significantly impact performance. Small LSH dimensions slightly outperform larger LSH dimensions.

| LSH Dim | BERTScore F1 | Rouge L | GPT4 Judge | Compression Ratio | Cache Memory (GB) |
|---|---|---|---|---|---|
| 4 | 0.8807 | 0.3974 | 4.3833 | 0.3728 | 2.8062 |
| 8 | 0.8802 | **0.3975** | **4.4113** | 0.3734 | 2.8355 |
| 16 | **0.8807** | 0.3972 | 4.3753 | 0.3716 | 2.8941 |
| 24 | 0.8802 | 0.3951 | 4.3733 | 0.3711 | 2.9527 |
| 32 | 0.8796 | 0.3926 | 4.3220 | 0.3710 | 3.0113 |
| 64 | 0.8797 | 0.3900 | 4.2333 | 0.3702 | 3.2456 |

# D    ATTENTION SCORES AND KEY NORMS VISUALIZATION

We further examine the method of our chief competitor, the $L_2$ eviction method (Devoto et al., 2024). In particular, in Figure 7 we examine the key-norm-attention correlation suggested by the authors. Indeed, low key-norms, even across prompts, demonstrate a strong correlation with attention score.

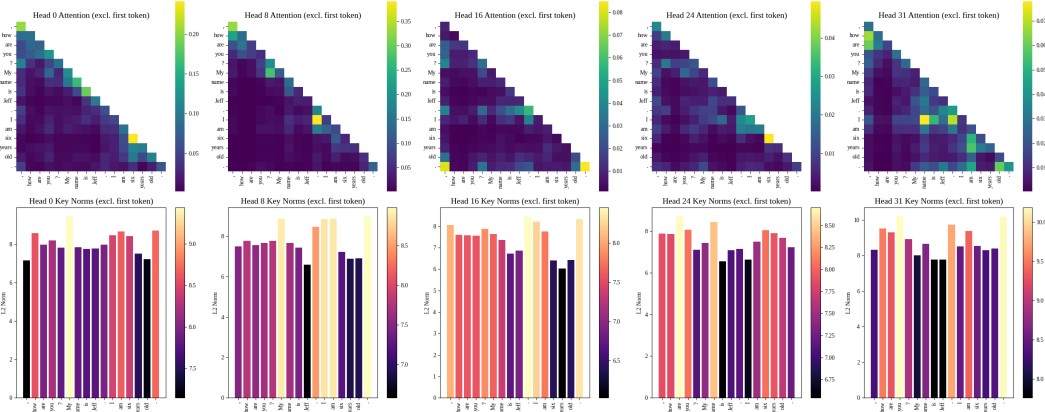

Figure 7: **Attention and Key Norms.** Attention scores and corresponding $L_2$ norms of key vectors (excluding the first token) for a sample of heads (0,8,16,24,31) in the 8th layer for a sample input sequence. Each subplot shows the attention heatmap (top) and the corresponding key norm values (bottom) for a particular head, allowing for a direct comparison between attention patterns and key norm values across different heads.

# E    ATTENTION LOSS RATIO ANALYSIS

We perform an attention loss ratio (ALR) analysis between LSH-based ranking and $L_2$-based ranking. Our implementation is an adaptation of the methodology described in Devoto et al. (2024). This section explores how much of the uncompressed attention matrix is preserved between LSH-E and the $L_2$ eviction strategy in Devoto et al. (2024).

Compressing the KV cache entails dropping KV pairs. Per (Devoto et al., 2024), we can define the attention loss caused by the compression as the sum of the attention scores associated with the dropped KV pairs in layer $l$ and head $h$ via the equation $L_{l,h}^m = \sum_{p \in D_{l,h}^m} a_{l,h,p}$, where $a_{l,h,p}$ is the average attention score at position $p$ for layer $l$ and head $h$, and $D_{l,h}^m$ denotes the positions of the $m$ dropped KV pairs, with $|D_{l,h}^m| = m$. We process a selection of prompts and examine how proposed evictions by the $L_2$ eviction strategy and LSH-E would affect the sum of attention scores.

To quantify the additional attention loss introduced by using an alternative ranking method (such as $L_2$ norm or LSH-E's $F_{score}$) instead of the true attention-based ranking, we define the cumulative attention loss difference as:

$$Y_{l,h} = \sum_{m=1}^{n} \left( L_{l,h}^m - L_{l,h,\text{ref}}^m \right),\tag{7}$$

where $L_{l,h,\text{ref}}^m$ is the cumulative attention loss when dropping the KV pairs with the actual lowest attention scores. The value $Y_{l,h}$ is non-negative, and a lower value indicates that the ranking method closely approximates non-compressed attention. Figure 8 depicts the ALR for the $L_2$ eviction rankings and an LSH ranking.

It is important to note that LSH-E is not designed to produce a global ranking among the keys as the $L_2$ method is designed to do (via a low-to-high ordering of all $L_2$ key norms). LSH-E ranks the importance of past tokens with regards to the current token – and this ranking changes every step. To simulate a comparison, we record the average Hamming distance between the key code of token $i$ and the query codes of all tokens $j > i$. We then sort tokens from lowest to highest average Hamming distance. Figure 8a reflects the ALR according to this ranking system. The $L_2$ ranking

exclusively prefers high-attention tokens, while the LSH ranking prefers medium-to-high-attention tokens. Based on our empirical results in Section 4, the selection of tokens over a spectrum of attention scores skewing towards high results in greater task versatility compared to the $L_2$ eviction.

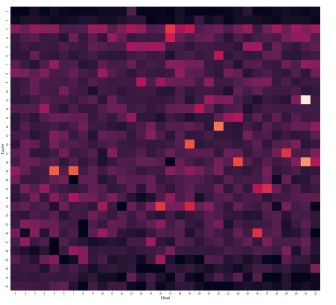 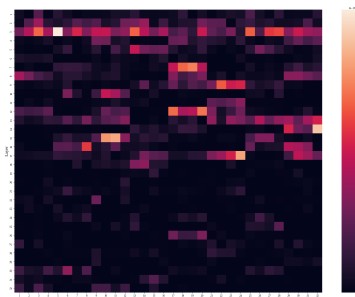

|  (a) ALR using LSH ranking  |  (b) ALR using $L_2$ ranking  |

Figure 8: **Attention Loss Ratio (ALR).** We compare how the eviction strategy of LSH-E and the $L_2$ method (Devoto et al., 2024) affects the ALR per equation 7. Our tested model is Llama3-8B-Instruct, which contains 32 heads and 32 attention layers. Cell $(i, j)$ depicts the ALR of head $i$ in attention layer $j$. A darker score indicates a lower ALR. The $L_2$ method exhibits extremely low ALR, thus indicating exclusive preference for high-attention tokens. LSH-E prefers to select medium-to-high attention tokens.

# F   ANALYSIS OF THE RELATIONSHIP BETWEEN ATTENTION SCORES AND LSH HAMMING DISTANCE

In this section, we follow up on our ALR in Appendix Section E. We analyze the relationship between attention scores and average LSH Hamming distances using 50 randomly selected prompts from GSM8K. We stress that this metric does not perfectly capture the "ranking" system of LSH-E (which cannot perform a global/full-sequence token-importance ranking like $L_2$ eviction).

For each prompt, we performed the following:

1. **Captured States**: Extracted normalized key and query vectors from every layer and head combination after applying rotary positional embeddings.

2. **Applied Random Projections**: Applied multiple random Gaussian projections, varying the projection length (number of bits). We tested with projection lengths of 8, 16, 24, and 32.

3. **Computed Hamming Distances**: Computed the Hamming distances between the projected and binarized vectors and averaged this over multiple projections to mitigate the randomness that LSH introduces and to obtain a more stable estimate of the Hamming distances.

4. **Computed Correlations**: Calculated the Pearson correlation coefficient between the attention scores and the inverted average Hamming distance for each layer and head combination and for each projection length.

## F.1   RESULTS

The average Pearson correlation between the attention scores and the inverted average Hamming distances is $0.2978 \pm 0.1947$. Table 10 and Figure 9a detail the average Pearson correlation per projection length.

## F.2   OBSERVATIONS

- **Correlation with Projection Length:** As shown in Figure 9a and Table 10 the average Pearson correlation increases with projection length. This is likely due to the more detailed

Table 10: Average Pearson correlation between attention scores and inverted average Hamming distances per projection length, computed for 50 randomly selected prompts from GSM8k. Higher projection lengths have stronger correlations.

| Projection Length | Mean | Standard Deviation |
|---|---|---|
| 8 | 0.2017 | 0.1890 |
| 16 | 0.2793 | 0.1852 |
| 24 | 0.3345 | 0.1806 |
| 32 | 0.3754 | 0.1792 |

vector representation in the projected space, allowing for finer-grained similarity comparisons.

- **Layer-wise Trends:** Figure 9b shows a slight decrease in the average Pearson correlation for the later transformer layers. Earlier layers may be more focused on recognizing broader patterns where the similarity LSH captures is more pronounced compared to the latter layers, which may focus on specifics not captured as effectively by Hamming distances.

- **Head-wise Consistency:** The correlation between attention scores and inverted average Hamming distance is relatively consistent across different attention heads, with little variance as seen in Figure '9c. This uniform behavior indicates that the relationship between attention scores and LSH-measured similarity is, to a large extent, independent of specific head functions.

- **LSH vs. $L_2$ Norms:** While $L_2$ norms were more effective at identifying high-attention tokes, LSH excelled at identifying tokens with moderate attention scores that are vital for the generation of coherent language output. This aligns with the findings of Guo et al. (2024), which suggests that tokens with low to medium attention scores are crucial for high-quality language generation.

- **LSH and Token Similarity:** LSH tended to group tokens together that are similar across dimensions, producing lower Hamming distances. Tokens with very high attention scores may only have strong associations for a relatively small subset of dimensions, which may not always be captured effectively by LSH.

## F.3 ALR COMPUTATION METHODOLOGY

We compute the Attention Loss Ratio (ALR) for each layer $l$ and head $h$ as follows:

1. **Data Capture** During the model's forward pass, we capture the necessary data for analysis:
   - **Attention Probabilities** $a_{l,h} \in \mathbb{R}^{n \times n}$: The attention scores between queries and keys.
   - **Key Norms** $\|\mathbf{k}_{l,h,p}\|_2$: The $L_2$ norms of key vectors at each position $p$.
   - **Key and Query Vectors** $\mathbf{k}_{l,h,p} \in \mathbb{R}^d$ and $\mathbf{q}_{l,h,p} \in \mathbb{R}^d$: Used for LSH ranking.

2. **Mean Attention Scores** For each token position $p$, we compute the mean attention score across all positions it attends to:

$$\bar{a}_{l,h,p} = \frac{1}{n} \sum_{q=1}^{n} a_{l,h,p,q}. \tag{8}$$

3. **Ranking Methods**
   - **Ideal Attention-Based Ranking** Rank positions in ascending order of $\bar{a}_{l,h,p}$ (from lowest to highest attention score).
   - $L_2$ **Norm Ranking** Rank positions in descending order of the key norms $\|\mathbf{k}_{l,h,p}\|_2$.
   - **LSH Ranking** Apply Locality-Sensitive Hashing (LSH) to key and query vectors using random projections, compute Hamming distances, and rank positions in ascending order of the average Hamming distance.

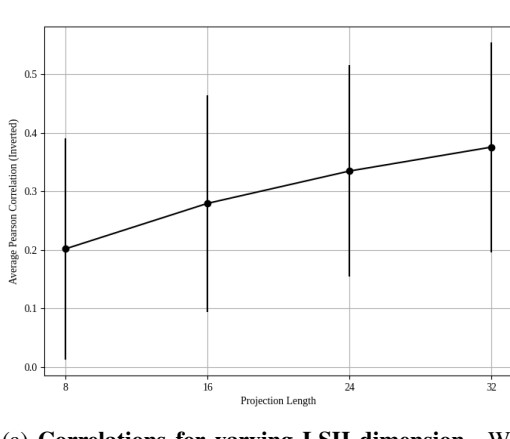

(a) **Correlations for varying LSH dimension.** We study the Pearson correlations between attention scores and the inverted average Hamming distances, computed over 50 randomly selected prompts from GSM8K, as a function of projection length for Llama-3-8B-Instruct. The tested projection lengths are 8,16, 24, and 32. The error bars indicate the standard deviation. Correlation strengthens as projection length increases.

(b) **Correlations by layer.** We measure the Pearson correlations between attention scores and the inverted average Hamming distances for each transformer layer in Llama-3-8B-Instruct computed over 50 randomly selected prompts from GSM8K. Error bars indicate standard deviation. The final three layers have the weakest correlations.

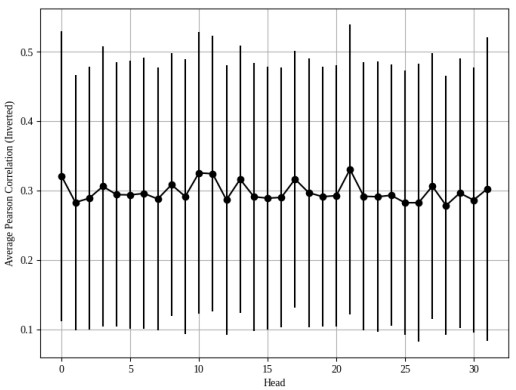

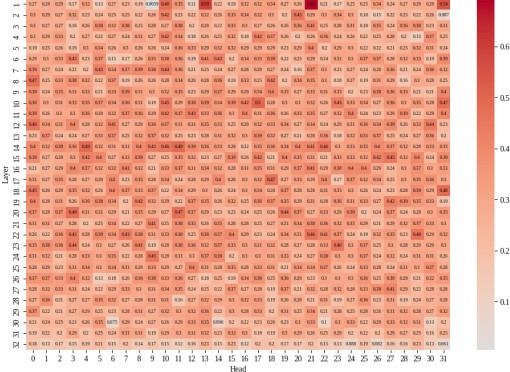

(c) **Correlations by head.** We study the Pearson correlation between attention scores and the inverted average Hamming distances for each head in Llama-3-8B-Instruct computed over 50 randomly selected prompts from GSM8K. Error bars indicate standard deviation. There is minimal variation between heads.

(d) **Correlation Heat Map.** We examine the average Pearson correlation between attention score and the inverted average Hamming distances (LSH ranking) across all layers and attention heads of Llama-3-8B-Instruct. As attention mass tends to concentrate over a few tokens (Gupta et al., 2021; Sheng et al., 2023), the slightly-weak, but positive correlation indicates the LSH ranking is selecting medium-to-high-attention tokens.

Figure 9: **Correlations of Attention and Inverted Hamming Distances**

4. **ALR Calculation** For each $m$ from 1 to $n$, compute the cumulative attention losses: This allows us to quantitatively compare how well different ranking methods (e.g., $L_2$ norm and LSH ranking) approximate the ideal scenario where the least important KV pairs (those with the lowest attention scores) are dropped during cache compression.

$$L_{l,h}^m = \sum_{i=1}^m \bar{a}_{l,h,\pi(i)}, \tag{9}$$

$$L_{l,h,\text{ref}}^m = \sum_{i=1}^m \bar{a}_{l,h,\sigma(i)}, \tag{10}$$

where $\pi(i)$ and $\sigma(i)$ are the indices of the $i$-th position in the ranking method and the ideal attention-based ranking, respectively. The ALR for each head and layer is then calculated as $Y_{l,h} = \sum_{m=1}^n \left( L_{l,h}^m - L_{l,h,\text{ref}}^m \right)$.

A lower $Y_{l,h}$ indicates that the ranking method closely approximates the ideal attention-based compression.

5. **Aggregation** We repeat the above steps for multiple prompts and average the ALR values to obtain the final ALR matrix across layers and heads.

## G METRICS AND PROMPTS

### G.1 STRING MATCH SCORE

The string matching score is calculated as:

$$\text{String Matching Score} = \frac{\text{Number of correctly matched characters in predicted string}}{\text{Total number of characters in GT}} \times 100$$

### G.2 GPT-4-JUDGE PROMPT

For the GPT-4-Judge metric used in evaluating free response question answering tasks, we accessed the GPT-4o model through OpenAI's API.

For the GPT4-Rouge metric, the prompt given to the model is:

```
You are shown ground-truth answer(s) and asked to judge the quality of an
    LLM-generated answer.
Assign it a score from 1-5 where 1 is the worst and 5 is the best based
    on how similar it is to the ground truth(s).
Do NOT explain your choice. Simply return a number from 1-5.

====GROUND TRUTHS====
{labels}

====ANSWER====
{prediction}
```

For the other three GPT4-Judge based on criteria, the prompt given to the model is:

```
You are shown a prompt and asked to assess the quality of an LLM-
    generated answer on the following dimensions:

===CRITERIA===
{criteria}

Respond with "criteria: score" for each criterion with a newline for each
    criterion.
Assign a score from 1-5 where 1 is the worst and 5 is the best based on
    how well the answer meets the criteria.

====PROMPT====
{prompt}

====ANSWER====
{prediction}
```

The list of criteria is:

```
CRITERIA = {
    "helpful": "The answer executes the action requested by the prompt
        without extraneous detail.",
    "coherent": "The answer is logically structured and coherent (ignore
        the prompt).",
    "faithful": "The answer is faithful to the prompt and does not contain
         false information.",
}
```

