# OpenReview forum: "LSH Tells You What To Discard: An Adaptive Locality-Sensitive Strategy for KV Cache Compression"
_ICLR.cc/2025/Conference — ICLR 2025 Conference Withdrawn Submission_

### Official Review · Reviewer_a2yh · 2024-11-04

**Soundness:** 2
**Presentation:** 3
**Contribution:** 3
**Rating:** 6
**Confidence:** 4

**Summary:**

This paper introduces a KV cache compression method based on LSH and shows that LSH-E can achieve good downstream performance on various downstream tasks with a 30%- 70% compression ratio.

**Strengths:**

This paper applies novel LSH methods to KV cache problems. The motivations and reasons why LSH can produce a good performance are well discussed. Besides this, a static compression rate of 30% - 70% is also helpful for many LLM serving systems, given the accuracy is preserved.

**Weaknesses:**

1. There is no comparison with other static KV compression baselines, including H2O, streamingLLM, and SnapKV. If this problem is solved, I will raise my score.
2. Only the memory compression ratio is shown. I will ask for the wall clock speedups (latency or throughput).

**Questions:**

Besides the problems mentioned in Weakness,
1 Does this method work well with quantization (KIVI, AWQ)?
2 How long does LSH-E increase first token latency?

These two questions can be left for future work.

---

> ### Author Response · Authors · 2024-11-28
> **Rebuttal by Authors**
>
> We thank the reviewer for the valuable feedback and suggestions. Below, we address all stated weaknesses and questions.
> ### Weakness 1
> > There is no comparison with other static KV compression baselines, including H2O, streamingLLM, and SnapKV.
>
> Thanks for this suggestion. We have added comparisons to several other well-cited KV cache compression strategies as baselines: H$_2$O [1], ScissorHands [2], and FastGen [3].  Our new results show that LSH-E performs comparably to H$_2$O and Scissorhands, and outperforms L2 and Fastgen on free form question answering.
>
> We have also included two new tasks from LongBench [4]: MultiNews and GovReport. Both are long-context summarization tasks since this task type was missing from our suite of evaluations. Per Tables 1 and 2 our method demonstrates comparable or superior performance on the two new LongBench tasks across various KV cache budgets. In the MultiNews summarization task, LSH-E achieves higher Rouge L score at most cache budgets, outperforming all baselines.
>
> ### Weakness 2
> > Show metrics for latency or throughput, not just compression ratio.
>
> Thanks for the suggestion. We have added throughput metrics on the LongBench GovReport summarization task in Table 4. LSH-E's prefill speed is **1.5-2x as fast as H$_2$O and Scissorhands** and **17x as fast** as FastGen even without low-level optimizations (i.e., expressing our hash tables in true binary bits). At the decoding stage, LSH-E is also comparable to L2 and faster than the other baseline methods.
>
> #### Table 4: Prefill and Decode Speed on LongBench MultiNewss Summarization
> | Strategy               | Cache Size       | Rouge L | Decode Toks Per Sec | Prefill Toks Per Sec |
> |------------------------|--------------------------------|---------|---------------------|-----------------------|
> | Full          | 100%  | 0.192   | 16.071  | 16573.492   |
> | LSH-E      | 30%   | 0.180   | 22.880  | 20293.524  |
> | L2         | 30%   | 0.165   | 23.981  | 20628.160   |
> | H$_2$O      | 30%    | 0.175   | 21.555  | 13025.776    |
> | Scissorhands     | 30%    | 0.175   | 21.448 | 13004.254   |
> | LSH-E    | 50%  | 0.186   | 22.846  | 20459.961  |
> | L2  | 50%   | 0.174   | 16.013  | 15851.952  |
> | H$_2$O     | 50%   | 0.181   | 21.973  | 13969.985   |
> | Scissorhands   | 50%  | 0.182   | 20.978  | 13549.967  |
> | LSH-E   | 70%  | 0.187   | 22.914   | 21002.334  |
> | L2     | 70%  | 0.187   | 24.305  | 21303.763 |
> | H$_2$O   | 70%   | 0.184   | 21.793 | 14050.521  |
> | Scissorhands    | 70%   | 0.183   | 21.705 | 13954.693  |
> | LSH-E    | 90%   | 0.185   | 22.873      | 21229.230      |
> | L2   | 90%   | 0.186   | 24.010    | 21305.693      |
> | H$_2$O  | 90%   | 0.181   | 21.665     | 14007.697     |
> | Scissorhands    | 90%     | 0.182   | 21.411   | 14025.440 |
> | Fastgen    | Attention recovery frac 70%   | 0.129   | 12.752     | 1171.069    |
> | Fastgen | Attention recovery frac 75%   | 0.174   | 12.291     | 1157.987   |
> | Fastgen   | Attention recovery frac 80%   | 0.184   | 11.850    | 1142.679      |
> | Fastgen    | Attention recovery frac 85%   | 0.183   | 11.658     | 1164.689     |
>
> ### Question 1
> > Does this method work well with quantization (KIVI, AWQ)?
>
> LSH-E will work with quantization. Additionally LSH and Simhash can also be used as a quantization method. Although we did not experiment with combining LSH-E and quantization, we think it will be a good inclusion in a future work.
>
> ### Question 2
> > How long does LSH-E increase first token latency?
>
> While we don't have specific numbers on Time-to-first-token (TTFT), our throughput results in Table 4 show that LSH-E is much faster at the pre-fill stage compared to attention-accumulation methods such as H$_2$O and Scissorhands and is on par with L2. Thus, the time to first token latency should be smaller than H$_2$O, Scissorhands and Fastgen and similar to that of L2.
>
> ---
> Thank you for your review. If we have addressed your questions, we would appreciate it if you would consider updating your score. If any other questions or concerns remain, please let us know.
>
> ### References
> [1] Zhang, Z., Sheng, Y., Zhou, T., Chen, T., Zheng, L., Cai, R., ... & Chen, B. (2023). H2o: Heavy-hitter oracle for efficient generative inference of large language models.
>
> [2] Liu, Z., Desai, A., Liao, F., Wang, W., Xie, V., Xu, Z., ... & Shrivastava, A. (2024). Scissorhands: Exploiting the persistence of importance hypothesis for llm kv cache compression at test time.
>
> [3] Ge, S., Zhang, Y., Liu, L., Zhang, M., Han, J., & Gao, J. (2023). Model tells you what to discard: Adaptive kv cache compression for llms.

---

> ### Author Response · Authors · 2024-12-01
>
> Dear Reviewer,
>
> We greatly appreciate your feedback. We have addressed your questions and concerns in our rebuttal. Please let us know if you have any further comments.
>
> Thank you,

---

> ### Comment · Reviewer_a2yh · 2024-12-01
> **Clarify for several problems**
>
> >> However, the size of the KV cache scales quadratically with sequence length n and linearly with
> the number of attention layers and heads. (Line 38-39)
>
> This is not true—the size of the KV cache scales linearly with sequence length n.
>
> >> For example, maintaining the KV cache for a sequence of 4K tokens in half-precision (FP16) can require approximately ∼16GB
> of memory for most models within the Llama 3 family (Dubey et al., 2024). (Line 42-43)
>
> This is also not true. 4K context length only occupies 500MB for Llama3-8B or 1.25GB for Llama3-70B or 2GB for Llama3-405B.
>
> Usually, when the context size is not very large, the majority of time is spent on MLP instead of attention. Typically, the boundary lies between 16 K and 32 K, depending on the model arch and GPUs.
>
> To make sure readers well understand the technique presented in the paper, I will ask for
> - 1. The average context length of the benchmark tested, especially the benchmark with the longest average context lengths.
> - 2. The GPU used in the experiments (and the framework, e.g., TensorRT-LLM, vLLM, SGLang, MLC-LLM, or native pytorch/Jax).
> - 3. Explain the two problems I mentioned above. For example, it can be a typo and how to modify them in the future version (I know the PDF deadline has passed). Other clarification (e.g., I (the reviewer) could also be wrong) is also acceptable.

---

> > ### Author Response · Authors · 2024-12-02
> > **Follow-up author response**
> >
> > Thank you for your follow-up feedback. We address them below.
> >
> > >> However, the size of the KV cache scales quadratically with sequence length n and linearly with the number of attention layers and heads. (Line 38-39)
> > > This is not true—the size of the KV cache scales linearly with sequence length n.
> >
> > This indeed was a typo which will be amended to reflect that the KV cache scales linearly with sequence length. This was likely spliced from text which indicated that attention without caching scales quadratically. It will be amended.
> >
> > >> For example, maintaining the KV cache for a sequence of 4K tokens in half-precision (FP16) can require approximately ∼16GB of memory for most models within the Llama 3 family (Dubey et al., 2024). (Line 42-43)
> > > This is also not true. 4K context length only occupies 500MB for Llama3-8B or 1.25GB for Llama3-70B or 2GB for Llama3-405B.
> >
> > This was a typo as well. It should have followed this formula (accounting for $K$ and $V$ matrices in the cache):
> >
> > **Total size of KV cache (bytes) = (batch size) * (sequence length) * 2 * (num layers) * (hidden size) *  sizeof(FP16)**
> >
> > Some example calculations: for a sequence length of 4000 this is approximately ~2 GB for Llama2-7B. For Llama3-8B, since it typically uses 8 attention heads for keys and values (via grouped-query attention) this results in the 500MB calculation that you mentioned. This formula is commonly used for rough estimation of memory complexity, for example, [on this NVIDIA guide.](https://developer.nvidia.com/blog/mastering-llm-techniques-inference-optimization/) This statement will be amended along with inclusion of this formula.
> >
> >
> >
> > > Usually, when the context size is not very large, the majority of time is spent on MLP instead of attention. Typically, the boundary lies between 16 K and 32 K, depending on the model arch and GPUs.
> >
> >
> > > To make sure readers well understand the technique presented in the paper, I will ask for
> >
> > > The average context length of the benchmark tested, especially the benchmark with the longest average context lengths.
> >
> > Thanks for the clarifying question. Please see the table below for the statistics on the long context-retrieval datasets used in this work. The tokenizer used is from the Llama3-8B-Instruct model.
> >
> > |Task|Number of Samples| Avg Prompt Tokens | Max Prompt Tokens | Min Prompt Tokens | Std Prompt Tokens |
> > |-----------|------|----------|-------|------|---------|
> > | Ruler Common Words | 500  | 3791.21  | 3980  | 3613 | 68.29   |
> > | Ruler Needle-In-A-Haystack | 500  | 3819.52  | 3831  | 3811 | 3.49    |
> > | LongBench MultiNews | 200  | 2650.11  | 13977 | 172  | 2133.29 |
> > | LongBench GovReport | 200  | 10286.41 | 51438 | 2065 | 6687.87 |
> >
> > > The GPU used in the experiments (and the framework, e.g., TensorRT-LLM, vLLM, SGLang, MLC-LLM, or native pytorch/Jax).
> >
> > We used an Nvidia H100 80GB for the two LongBench summarization tasks (GovReport and MultiNews) and an Nvidia L4 for all other tasks. We used cold-compress as our testing framework, which is implemented in PyTorch. These benchmark statistics will be added to our next revision within the Appendix and referenced within our "Experiments" section.

---

> > > ### Comment · Reviewer_a2yh · 2024-12-02
> > >
> > > Thanks for your response. I wonder why H2O suffers from a decrease in throughput compared to full attention.

---

> > > > ### Author Response · Authors · 2024-12-03
> > > >
> > > > > I wonder why H2O suffers from a decrease in throughput compared to full attention.
> > > >
> > > > Both H2O and Scissorhands have lower throughput compared to full attention because of the overhead introduced by attention accumulation or attention averaging (Scissorhands). This is made more obvious by the very long prompts in the MultiNews task.

---

> > > > > ### Comment · Reviewer_a2yh · 2024-12-03
> > > > >
> > > > > I disagree with the measured throughput. Even if H2O requires accumulating attention scores, a 40% decrease in performance is impossible.
> > > > >
> > > > > Personally, I guess the difference may lie in full attention can use flash-attn, but H2O does not.
> > > > >
> > > > > However, other than this, I think most concerns are addressed. The typos are fixed and the baselines are presented. I decide to raise my score to 6.

---

### Official Review · Reviewer_rWSu · 2024-11-04

**Soundness:** 2
**Presentation:** 1
**Contribution:** 2
**Rating:** 3
**Confidence:** 4

**Summary:**

This paper proposes a method that uses LSH to perform kv cache eviction. The provided experiments show that the proposed method outperforms the baseline.

**Strengths:**

Strong Points
----
S1. The problem of the paper is well-motivated.

S2. The proposed algorithm is simple and clear with illustrative example.

S3. The proposed method outperforms the baseline L2.

**Weaknesses:**

Weak Points
----
W1. Important related studies and baselines are missing:
Singhania, P., Singh, S., He, S., Feizi, S., & Bhatele, A. (2024). Loki: Low-Rank Keys for Efficient Sparse Attention. arXiv preprint arXiv:2406.02542.
Tang, J., Zhao, Y., Zhu, K., Xiao, G., Kasikci, B., & Han, S. (2024). Quest: Query-Aware Sparsity for Efficient Long-Context LLM Inference. arXiv preprint arXiv:2406.10774.

W2. The key measures of the targeted task should be have more accurate inference with lower memory footprint and latency. I do not agree with the methodology of not comparing with other "non attention-free" methods.

W3. The presentation of experiments need to be improved: Lack of discussions and intuitions in the experiment analysis. For example, why does LSH-E outperform Full in Figure 4a; why does LSH-E become worse than L2 after 50% cache budget in Figure 4b? We have many subsubsections in the experiments, but most contents in those are barely text illustration of the figure and result while no discussion of why we would have those results.

W4. The execution time of the proposed system is missing.

W5. The discussion of the error introduced by the LSH is not included. I wonder what if we use cosine similarity to evict the cache instead of LSH, how will be the accuracy, latency, and memory usage?

W6. In the supplementary materials, we see more experiments with more baselines that are better than L2. I wonder the reason why the authors do not include them.


Presentation
----
P1. Line 180 "heavy hitters' -> ``heavy hitters''
P2. The axis captions of the figures are too small to be seen.

**Questions:**

See weakness.

---

> ### Author Response · Authors · 2024-11-28
> **Rebuttal by Authors Pt 1**
>
> We thank the reviewer for the valuable feedback and suggestions. We are encouraged that you find our work simple, clear and well-motivated. Below, we address all stated weaknesses and questions.
>
> ### Weakness 1
> > Important related studies and baselines are missing: Singhania, P., Singh, S., He, S., Feizi, S., & Bhatele, A. (2024). Loki: Low-Rank Keys for Efficient Sparse Attention. arXiv preprint arXiv:2406.02542. Tang, J., Zhao, Y., Zhu, K., Xiao, G., Kasikci, B., & Han, S. (2024). Quest: Query-Aware Sparsity for Efficient Long-Context LLM Inference. arXiv preprint arXiv:2406.10774.
>
> Thank you for suggesting these baselines. Respectfully, we disagree that these methods demonstrate significant overlap with our approach as they are attention efficiency / approximation methods, rather than KV cache eviction strategies.
>
> ### Weakness 2 & 4
> > The key measures of the targeted task should be have more accurate inference with lower memory footprint and latency. I do not agree with the methodology of not comparing with other "non attention-free" methods.
>
> > The execution time of the proposed system is missing.
>
> Thank you for your suggestion. We have added comparisons to several other well-cited KV cache compression strategies as baselines: H2O [1], ScissorHands [2], and FastGen [3]. We have updated existing experiments in the paper to include these new baselines.
>
> We also included two additional tasks from LongBench [4]: MultiNews and GovReport. Both are long-context summarization tasks, since this task type was missing from our suite of evaluations. Additionally we have added pre-fill and decoding speed metrics on the LongBench MultiNews dataset.
>
> Our new results show that LSH-E performs comparably to H2O and Scissorhands, and outperforms L2 and Fastgen on free form question answering tasks. In the two new summarization tasks, LSH-E consistently demonstrates comparable or superior Rouge L scores across various cache budgets. In the MultiNews summarization task, LSH-E achieves higher Rouge L score at most cache budgets, outperforming all baselines, demonstrating LSH-E’s robustness and effectiveness in handling very large context lengths. LSH-E is also faster: our pre-fill stage is 1.5-2x as fast as attention-dependent methods like H2O and Scissorhands, and 17x as fast compared to FastGen. At the decoding stage LSH-E is also comparable to L2 and faster than the other baseline methods. Please see table below for more details.
>
> ### Table Results of LongBench GovReport and MultiNews Summarization with Throughput
> |  |  | GovReport | MultiNews |  |  |
> |---|---|---|---|---|---|
> | Strategy | Cache Budget | Rouge L | Rouge L | Decode Toks Per Sec | Prefill Toks Per Sec |
> | Full | 100% | 0.230 | 0.192 | 16.071 | 16573.492 |
> | LSH-E | 30% | 0.202 | 0.180 | 22.880 | 20293.524 |
> | L2 | 30% | 0.201 | 0.165 | 23.981 | 20628.160 |
> | H2O | 30% | 0.219 | 0.175 | 21.555 | 13025.776 |
> | Scissorhands | 30% | 0.214 | 0.175 | 21.448 | 13004.254 |
> | LSH-E | 50% | 0.217 | 0.186 | 22.846 | 20459.961 |
> | L2 | 50% | 0.214 | 0.174 | 16.013 | 15851.952 |
> | H2O | 50% | 0.225 | 0.181 | 21.973 | 13969.985 |
> | Scissorhands | 50% | 0.219 | 0.182 | 20.978 | 13549.967 |
> | LSH-E | 70% | 0.223 | 0.187 | 22.914 | 21002.334 |
> | L2 | 70% | 0.223 | 0.187 | 24.305 | 21303.763 |
> | H2O | 70% | 0.229 | 0.184 | 21.793 | 14050.521 |
> | Scissorhands | 70% | 0.226 | 0.183 | 21.705 | 13954.693 |
> | LSH-E | 90% | 0.228 | 0.185 | 22.873 | 21229.230 |
> | L2 | 90% | 0.230 | 0.186 | 24.010 | 21305.693 |
> | H2O | 90% | 0.227 | 0.181 | 21.665 | 14007.697 |
> | Scissorhands | 90% | 0.230 | 0.182 | 21.411 | 14025.440 |
> | Fastgen | Attention recovery frac 70% | 0.192 | 0.129 | 12.752 | 1171.069 |
> | Fastgen | Attention recovery frac 75% | 0.231 | 0.174 | 12.291 | 1157.987 |
> | Fastgen | Attention recovery frac 80% | 0.232 | 0.184 | 11.850 | 1142.679 |
> | Fastgen | Attention recovery frac 85% | 0.236 | 0.183 | 11.658 | 1164.689 |
>
> ### Weakness 3
> > The presentation of experiments need to be improved: Lack of discussions and intuitions in the experiment analysis. For example, why does LSH-E outperform Full in Figure 4a; why does LSH-E become worse than L2 after 50% cache budget in Figure 4b? We have many subsubsections in the experiments, but most contents in those are barely text illustration of the figure and result while no discussion of why we would have those results.
>
> Thank you for your suggestion. We will update the paper to include more analysis and discussions of experiment results.
> KV cache eviction strategies sometimes perform better than using full cache because the evicted token is not always useful. Evicting useless tokens could actually help with the language quality of generated answers.

---

> ### Author Response · Authors · 2024-11-28
> **Rebuttal by Authors Pt 2**
>
> ### Weakness 5
> > The discussion of the error introduced by the LSH is not included. I wonder what if we use cosine similarity to evict the cache instead of LSH, how will be the accuracy, latency, and memory usage?
>
> We conducted attention loss analysis which approximates this error. Since our LSH projection is simply searching for large/small dot products, eviction via true cosine similarity would essentially be equivalent to conducting full attention with everything in the KV cache and removing the token with lowest attention score. It would be better to leverage a technique such as H$_2$O or ScissorHands which relies on accumulated attention in this scenario. In any case, it would result in $\mathcal{O}(N^2)$ memory and $\mathcal{O}(dN^2)$ computional complexity, where $d$ is the projection dimension, for a KV cache with $N$ tokens and worse latency due to the dot product calculation between floating-point vectors versus bit-wise comparison of Boolean hashes. Please let us know if this is not clear.
>
> Below is an experiment of attention loss for LSH-E, L2 and Scissorhands, quantifying the discrepancy introduced by the eviction strategy compared to maintaining the full cache. We measured the atttention loss of each attention head and report the average. Attention loss is defined as the sum of the attention probabilities for evicted tokens. Or equivalently, 1 - the sum of the attention probabilities for the tokens in the compressed cache.
>
> The attention loss was measured at 50% cache budget using prompts from the GSM8K question answering dataset. As per Table 5, all three methods have low attention loss at 50% cache budget, and LSH-E has lower attention loss compared to L2 and scissorhands, proving LSH-E's ability of keeping high attention tokens in the KV cache.  By quantifying attention loss, we demonstrated that LSH-E introduces minimal deviation from full-cache attention.
>
> ### Table: Attention Loss
> | Strategy| Attention Loss |
> |-----|------|
> | LSH-E | 0.03357896805 |
> | L2 | 0.03403072357 |
> | Scissorhands | 0.04483547211 |
>
>
> ### Weakness 6
> > In the supplementary materials, we see more experiments with more baselines that are better than L2. I wonder the reason why the authors do not include them.
>
> We calculated multiple metrics from the same family. For example we calculated four different variations of Rouge: Rouge 1, 2, L and Lsum, and preicions, recall and F1 of BertScore. We also used GPT4 as a judge on four different metrics: similiarity to the ground truth, helpfulness, coherentness and faithfulness. LSH-E outperforms the baslines on all these metrics on most of the experiments. But due to page limitations we chose to show only the metrics that are most relevant to each task / dataset in the paper.
>
>
> ### Presentation 1
> > Line 180 "heavy hitters' -> ``heavy hitters'' P2. The axis captions of the figures are too small to be seen.
>
> Thank you for pointing the typo. It has been fixed in the paper. We have also updated the figures to make the axis labels larger.
>
> ---
>
> Thank you for your review. If we have addressed your questions, we would appreciate it if you would consider updating your score. If any other questions or concerns remain, please let us know.
>
>
> ## Reference
>
> [1] Zhang, Z., Sheng, Y., Zhou, T., Chen, T., Zheng, L., Cai, R., ... & Chen, B. (2023). H2o: Heavy-hitter oracle for efficient generative inference [...].
>
> [2] Liu, Z., Desai, A., Liao, F., Wang, W., Xie, V., Xu, Z., ... & Shrivastava, A. (2024). Scissorhands: Exploiting the persistence [...]
>
> [3] Ge, S., Zhang, Y., Liu, L., Zhang, M., Han, J., & Gao, J. (2023). Model tells you what to discard: Adaptive kv cache compression for llms.
>
> [4] Bai, Y., Lv, X., Zhang, J., Lyu, H., Tang, J., Huang, Z., ... & Li, J. (2023). Longbench: A bilingual, multitask benchmark for long context understanding.

---

> ### Author Response · Authors · 2024-12-01
>
> Dear Reviewer,
>
> We greatly appreciate your feedback. We have addressed your questions and concerns in our rebuttal. Please let us know if you have any further comments.
>
> Thank you.

---

### Official Review · Reviewer_NvGH · 2024-11-04

**Soundness:** 2
**Presentation:** 1
**Contribution:** 1
**Rating:** 1
**Confidence:** 4

**Summary:**

This paper introduces LSH-E, an algorithm for compressing the key-value (KV) cache in large language models (LLMs) using locality-sensitive hashing (LSH). Despite the availability of prior work—including KDEformer, Hyperattention, SubGen, and QJL—that similarly utilizes LSH for efficient attention and memory management, these related efforts are not cited here. LSH-E leverages Hamming distance calculations in a binary space following a Quantized Johnson-Lindenstrauss (JL) transform (SimHash) to identify and evict tokens with low relevance to the current query, resulting in memory savings. This pre-attention approach provides a lightweight, GPU-efficient solution for long-context tasks, although its effectiveness ultimately depends on the algorithm’s CUDA implementation efficiency.

**Strengths:**

The use of theoretical approaches such as SimHash, a highly efficient hashing method, is a valuable aspect of this work, contributing to both the effectiveness and scalability of the proposed method.

**Weaknesses:**

- The term "novel" should not be used for LSH in this context, as it is not a new approach and has appeared in prior work. Specifically, the methods used in KDEformer, Hyperattention, QJL, and SubGen demonstrate significant overlap, yet these works are not cited here, despite their relevance.

- The experimental setup lacks comprehensiveness; comparisons with alternative methods like H2O, SubGen, and other established baselines should be included to provide a more robust evaluation.

- The datasets used in the experiments are not sufficiently large for evaluating performance in long-context scenarios. Given that these methods target long-sequence processing, experiments should ideally use token sizes over 50,000. LongBench or other large-scale datasets would be more appropriate for a thorough evaluation.

- Additionally, runtime metrics should be reported to assess the efficiency of token generation and to substantiate the computational benefits claimed in the paper.

KDEformer : https://proceedings.mlr.press/v202/zandieh23a.html
HyperAttention : https://arxiv.org/abs/2310.05869
SubGen :  https://arxiv.org/abs/2402.06082
QJL : https://arxiv.org/abs/2406.03482

**Questions:**

- Could you provide a plot showing the distortion error introduced by LSH compression across different levels of compression? Specifically, how does the approximation quality change as more tokens are evicted or as the quantization parameters are adjusted?

- Given that LSH-E’s efficiency largely depends on its CUDA implementation, can you elaborate on any specific optimizations made within the CUDA code?

- Could you clarify how LSH-E handles multi-head attention? Specifically, is each head processed separately with its own LSH compression, or is there a shared mechanism across heads?

---

> ### Author Response · Authors · 2024-11-28
> **Rebuttal by Authors Pt 1**
>
> We thank the reviewer for the valuable feedback and suggestions. Below, we address all stated weaknesses and questions. Given that a score of "1" is typically reserved for a work which is severely technically flawed or extremely incremental, if you believe that we have addressed your concerns, we would appreciate if the reviewer would be willing to reassess their score. We are more than happy to discuss any further concerns.
>
> ### Weakness 1
>
> > The term "novel" should not be used for LSH in this context, as it is not a new approach and has appeared in prior work. Specifically, the methods used in KDEformer, Hyperattention, QJL, and SubGen demonstrate significant overlap, yet these works are not cited here, despite their relevance.
>
> Respectfully, we disagree that these methods demonstrate significant overlap with our approach as it appears that only SubGen [5] is a token eviction strategy. It relies on reducing the cache by instead clustering embedding and choosing representatives from key clusters to process attention. It appears as though it must initially view all embeddings to perform this clustering, which is suitable for the CPU, but would result in VRAM blowup on the GPU for long enough context. In contrast, our approach simply looks at portion of the context within the memory budget to form an initial eviction and then proceeds token-by-token to swap embeddings in and out of the cache based purely on Hamming distances.
>
> Hyperattention [2], QJL [3], and KDEFormer [4] are using LSH to approximate the attention module $A$, apparently without token eviction, which is instead in the vein of works descending from Reformer [1]. However, all methods do appreciate memory-reductive effects, so we appreciate the reviewer pointing us towards this literature which have now been included in our related works discussion under "Memory-Efficient Transformers."
>
> ### Weakness 2 - 4
> > The experimental setup lacks comprehensiveness; comparisons with alternative methods like H2O, SubGen, and other established baselines should be included to provide a more robust evaluation.
>
> > The datasets used in the experiments are not sufficiently large for evaluating performance in long-context scenarios. Given that these methods target long-sequence processing, experiments should ideally use token sizes over 50,000. LongBench or other large-scale datasets would be more appropriate for a thorough evaluation.
>
> > Additionally, runtime metrics should be reported to assess the efficiency of token generation and to substantiate the computational benefits claimed in the paper.
>
> Thank you for your suggestion. We have added comparisons to several other well-cited KV cache compression strategies as baselines: H$_2$O [6], ScissorHands [7], and FastGen [9]. We have updated existing experiments in the paper to include these new baselines.
>
> We also included two additional tasks from LongBench [8]: MultiNews and GovReport. Both are long-context summarization tasks, since this task type was missing from our suite of evaluations. Additionally we have added pre-fill and decoding speed metrics on the LongBench MultiNews dataset.
>
> Our new results show that LSH-E performs comparably to H2O and Scissorhands, and outperforms L2 and Fastgen on free form question answering tasks. In the two new summarization tasks, LSH-E consistently demonstrates comparable or superior Rouge L scores across various cache budgets. In the MultiNews summarization task, LSH-E achieves higher Rouge L score at most cache budgets, outperforming all baselines, demonstrating LSH-E’s robustness and effectiveness in handling very large context lengths. LSH-E is also faster: our pre-fill is **1.5-2x as fast** as attention-dependent methods like H2O and Scissorhands, and **17x as fast** compared to FastGen. At the decoding stage LSH-E is also comparable to L2 and faster than the other baseline methods. Please see table below for more details.
>
>
> ### Question 2
> > Given that LSH-E’s efficiency largely depends on its CUDA implementation, can you elaborate on any specific optimizations made within the CUDA code?
>
> Despite that we have not used any CUDA optimization yet, LSH-E is already demonstrating comparable and even superior computational speed and memory efficiency compared to baseline methods. If we use actual bits for the LSH hash code, we can reduce the memory overhead of LSH-E by a factor of 8. We also expect faster hamming distance computation, thus increasing the throughput of LSH-E further.
>
> ### Question 3
> > Could you clarify how LSH-E handles multi-head attention? Specifically, is each head processed separately with its own LSH compression, or is there a shared mechanism across heads?
>
> Each head maintains its own LSH hash table and processes its own LSH compression and eviction.

---

> ### Author Response · Authors · 2024-11-28
> **Rebuttal by Authors Pt 2**
>
> ### Table 1: Results of LongBench GovReport and MultiNews Summarization with Throughput
> |  |  | GovReport | MultiNews |  |  |
> |---|---|---|---|---|---|
> | Strategy | Cache Budget | Rouge L | Rouge L | Decode Toks Per Sec | Prefill Toks Per Sec |
> | Full | 100% | 0.230 | 0.192 | 16.071 | 16573.492 |
> | LSH-E | 30% | 0.202 | 0.180 | 22.880 | 20293.524 |
> | L2 | 30% | 0.201 | 0.165 | 23.981 | 20628.160 |
> | H2O | 30% | 0.219 | 0.175 | 21.555 | 13025.776 |
> | Scissorhands | 30% | 0.214 | 0.175 | 21.448 | 13004.254 |
> | LSH-E | 50% | 0.217 | 0.186 | 22.846 | 20459.961 |
> | L2 | 50% | 0.214 | 0.174 | 16.013 | 15851.952 |
> | H2O | 50% | 0.225 | 0.181 | 21.973 | 13969.985 |
> | Scissorhands | 50% | 0.219 | 0.182 | 20.978 | 13549.967 |
> | LSH-E | 70% | 0.223 | 0.187 | 22.914 | 21002.334 |
> | L2 | 70% | 0.223 | 0.187 | 24.305 | 21303.763 |
> | H2O | 70% | 0.229 | 0.184 | 21.793 | 14050.521 |
> | Scissorhands | 70% | 0.226 | 0.183 | 21.705 | 13954.693 |
> | LSH-E | 90% | 0.228 | 0.185 | 22.873 | 21229.230 |
> | L2 | 90% | 0.230 | 0.186 | 24.010 | 21305.693 |
> | H2O | 90% | 0.227 | 0.181 | 21.665 | 14007.697 |
> | Scissorhands | 90% | 0.230 | 0.182 | 21.411 | 14025.440 |
> | Fastgen | Attention recovery frac 70% | 0.192 | 0.129 | 12.752 | 1171.069 |
> | Fastgen | Attention recovery frac 75% | 0.231 | 0.174 | 12.291 | 1157.987 |
> | Fastgen | Attention recovery frac 80% | 0.232 | 0.184 | 11.850 | 1142.679 |
> | Fastgen | Attention recovery frac 85% | 0.236 | 0.183 | 11.658 | 1164.689 |
>
>
> ---
> Thank you for your review. If we have addressed your questions, we would appreciate it if you would consider updating your score. If any other questions or concerns remain, please let us know.
>
>
> [1] Kitaev, N., Kaiser, Ł., & Levskaya, A. (2020). Reformer: The efficient transformer.
>
> [2] Han, I., Jayaram, R., Karbasi, A., Mirrokni, V., Woodruff, D. P., & Zandieh, A. (2023). Hyperattention: Long-context attention in near-linear time.
>
> [3] Zandieh, A., Daliri, M., & Han, I. (2024). QJL: 1-Bit Quantized JL Transform for KV Cache Quantization with Zero Overhead.
>
> [4] Zandieh, A., Han, I., Daliri, M., & Karbasi, A. (2023, July). Kdeformer: Accelerating transformers via kernel density estimation.
>
> [5] Zandieh, A., Han, I., Mirrokni, V., & Karbasi, A. (2024). SubGen: Token Generation in Sublinear Time and Memory.
>
> [6] Zhang, Z., Sheng, Y., Zhou, T., Chen, T., Zheng, L., Cai, R., ... & Chen, B. (2023). H2o: Heavy-hitter oracle for efficient generative inference [...].
>
> [7] Liu, Z., Desai, A., Liao, F., Wang, W., Xie, V., Xu, Z., ... & Shrivastava, A. (2024). Scissorhands: Exploiting the persistence [...]
>
> [8] Bai, Y., Lv, X., Zhang, J., Lyu, H., Tang, J., Huang, Z., ... & Li, J. (2023). Longbench: A bilingual, multitask benchmark for long context understanding.
>
> [9] Ge, S., Zhang, Y., Liu, L., Zhang, M., Han, J., & Gao, J. (2023). Model tells you what to discard: Adaptive kv cache compression for llms.

---

> ### Author Response · Authors · 2024-12-01
>
> Dear Reviewer,
>
> We greatly appreciate your feedback. We have addressed your questions and concerns in our rebuttal. Please let us know if you have any further comments.
>
> Thank you.

---

### Official Review · Reviewer_yqyi · 2024-11-04

**Soundness:** 2
**Presentation:** 3
**Contribution:** 2
**Rating:** 5
**Confidence:** 4

**Summary:**

This paper presents new methods to accelerate inference of auto-regressive transformers used in most modern-day decoder-based LLM architectures. Indeed, the main drawback of existing systems is the size of the "KV  Cache" or Key-Value Cache which is used during the attention mechanism. To speed up the attention calculation, most systems have a cache which remembers the keys and values of commonly used tokens, to avoid recomputing it for each token decoding.  However ,such a cache, for it to be performant at inference time, must scale quadratically with the sequence length, and linear in number of layers and attention heads.

(Authors: please explain why for the uninformed reader -- this is stated in the intro, but without explanation)

In this paper, the authors present an LSH based method to evict far-away key tokens. Indeed, suppose we have an LSH which gets a binary encoding of any vector using random hyperplane projection method (SIMHASH).
Then, we can first pre-process and compute the hamming distance between query token and all key tokens, and evict the farthest one, as this is the one least likely to affect the overall attention soft-max operation.

They implement this simple scheme and provide a range of quality vs cache size metrics comparing with one other KV-cache called L2-Dropout Cache, which drops the keys based on their magnitudes.

**Strengths:**

Studies an important problem of much significance in todays LLM era.

Presents a simple yet elegant approach

Does good evaluations on a range of use-cases

**Weaknesses:**

Why is there no timing experiment, since that will be one key benefit of caching.

Why only restrict to attention-free cache policies and specifically only compare with the L2-dropout baseline?

Conceptually, what is the key difference with Reformer? I have not read that paper but you mention in passing that it is using LSH and simhash also. Is which cells to evaluate vs what to evict the only difference between Reformer and your work? If so, worth comparing with Reformer also in plots?

What is the rationale of the policy? Why can't a token just evicted become relevant again? I guess is there some language-based "locality of reference"?

Do ablation of the hardcoded bits, i.e., you mention you hard-cache the first few and last few tokens. What is the contribution of this to your overall success metrics?

The eviction policy is not clearly understandable in how it aggregates the hamming distances over time steps. Is it only based on the most recent time step, or some more complex rule?

**Questions:**

Line 52: "However, this L2 dropout strategy only performs well on
long-context retrieval tasks. It is specialized to retain only those tokens with the highest attention" -- be more specific. Why is this?

Line 57: "wide variety of tasks?" -- how do you define this?

Line 145: Formally for our setup, distd(x, y) cos θx,y, here it is more a measure of cosine similarity than distance. Misleading, perhaps?

Line 419: did you mean "LSH dimension does significantly impact performance" --> does not?

---

> ### Author Response · Authors · 2024-11-28
> **Rebuttal by Authors Pt 1**
>
> We thank the reviewer for the valuable feedback and suggestions. Below, we address all stated weaknesses and questions.
>
> > However ,such a cache, for it to be performant at inference time, must scale quadratically with the sequence length, and linear in number of layers and attention heads.
> > (Authors: please explain why for the uninformed reader -- this is stated in the intro, but without explanation)
>
> Attention-based eviction strategies like H2O and Scissorhands needs to accumulate attention of each token in the KV cache. Assuming the size of the KV cache is N tokens, for each decoded token, N attention scores need to be added which requires $\mathcal{O}(N^2)$ computation (all pairwise dot-products) and storage ($N^2$ entries). Therefore the time complexity of maintaining accumulated attention of tokens is approximately O(N^2) or quadratic to the sequence length because N is a percentage of the max sequence length.
>
> ### Weakness 1 & 2
> > Why only restrict to attention-free cache policies and specifically only compare with the L2-dropout baseline?
>
> > Why is there no timing experiment, since that will be one key benefit of caching.
>
> Thanks for the suggestion. We have added comparisons to three more baselines: H2O, Scissorhands, and FastGen to contextualize LSH-E's performance against state-of-the-art methods. We have updated existing experiments in the paper to include these new baselines. Additionally, we added two long-context summarization tasks from the LongBench benchmarks: MultiNews and GovReport, and report results in the table below.
>
> In these new experiments, LSH-E consistently demonstrats comparable or superior Rouge L scores across various cache budgets. In the MultiNews summarization task, LSH-E achieves higher Rouge L score at most cache budgets, outperforming all baselines, demonstrating LSH-E’s robustness and effectiveness in handling very large context lengths.
>
> We also added timing experiments and report throughput metrics. We provide decoding and pre-fill tokens per second results on the LongBench MultiNews task. LSH-E is 1.5-2x as fast as H2O and Scissorhands, and 17x as fast as FastGen at the pre-fill stage. Even without low-level optimizations (e.g., expressing hash tables in binary bits), LSH-E proved to be as fast as the L2 strategy in decoding and significantly faster than attention-based baselines. This speedup was achieved while maintaining competitive quality metrics, demonstrating the computational efficiency of LSH-E.
>
> ### Table: Results of LongBench GovReport and MultiNews Summarization with Throughput
> |  |  | GovReport | MultiNews |  |  |
> |---|---|---|---|---|---|
> | Strategy | Cache Budget | Rouge L | Rouge L | Decode Toks Per Sec | Prefill Toks Per Sec |
> | Full | 100% | 0.230 | 0.192 | 16.071 | 16573.492 |
> | LSH-E | 30% | 0.202 | 0.180 | 22.880 | 20293.524 |
> | L2 | 30% | 0.201 | 0.165 | 23.981 | 20628.160 |
> | H2O | 30% | 0.219 | 0.175 | 21.555 | 13025.776 |
> | Scissorhands | 30% | 0.214 | 0.175 | 21.448 | 13004.254 |
> | LSH-E | 50% | 0.217 | 0.186 | 22.846 | 20459.961 |
> | L2 | 50% | 0.214 | 0.174 | 16.013 | 15851.952 |
> | H2O | 50% | 0.225 | 0.181 | 21.973 | 13969.985 |
> | Scissorhands | 50% | 0.219 | 0.182 | 20.978 | 13549.967 |
> | LSH-E | 70% | 0.223 | 0.187 | 22.914 | 21002.334 |
> | L2 | 70% | 0.223 | 0.187 | 24.305 | 21303.763 |
> | H2O | 70% | 0.229 | 0.184 | 21.793 | 14050.521 |
> | Scissorhands | 70% | 0.226 | 0.183 | 21.705 | 13954.693 |
> | LSH-E | 90% | 0.228 | 0.185 | 22.873 | 21229.230 |
> | L2 | 90% | 0.230 | 0.186 | 24.010 | 21305.693 |
> | H2O | 90% | 0.227 | 0.181 | 21.665 | 14007.697 |
> | Scissorhands | 90% | 0.230 | 0.182 | 21.411 | 14025.440 |
> | Fastgen | Attention recovery frac 70% | 0.192 | 0.129 | 12.752 | 1171.069 |
> | Fastgen | Attention recovery frac 75% | 0.231 | 0.174 | 12.291 | 1157.987 |
> | Fastgen | Attention recovery frac 80% | 0.232 | 0.184 | 11.850 | 1142.679 |
> | Fastgen | Attention recovery frac 85% | 0.236 | 0.183 | 11.658 | 1164.689 |
>
> ### Weakness 3
> > Conceptually, what is the key difference with Reformer? I have not read that paper but you mention in passing that it is using LSH and simhash also. Is which cells to evaluate vs what to evict the only difference between Reformer and your work? If so, worth comparing with Reformer also in plots?
>
> Thanks for the question. We clarified the conceptual distinctions between LSH-E and related works such as Reformer, H2O, and SubGen and updated the related works section of our paper.
>
> The biggest difference is that Reformer is an efficient attention replacement rather than a kv cache eviction strategy. Reformer and our work use the same tools but for different purposes and to achieve different goals. Reformer uses LSH and simhash to group tokens that are similar into buckets, and restrict attention computation to tokens within the same bucket for efficiency of computation. Our work uses LSH to find the least similar tokens in history and evict them from the KV cache for efficiency of memory usage.

---

> ### Author Response · Authors · 2024-11-28
> **Rebuttal by Authors Pt 2**
>
> ### Weakness 4
> > What is the rationale of the policy? Why can't a token just evicted become relevant again? I guess is there some language-based "locality of reference"?
>
> A token could become relevant again. But with the restriction of GPU memory and KV cache budget in mind, KV cache strategies must trade-off between the information lost and memory requirement. We have demonstrated through experimentst that LSH-E achieves good performance on multiple real-world tasks and better speed compared to attention-based eviction methods.
> [name=Furong: talking points. 1. there is evidence certain tokens are not crucial and evicting them do not hurt performance significantly. cite. 2. empirically, our results verify this. some of your discussion above can be moved here. 3. certain tasks require more dynamically changing tokens, while certain tasks do not. we will leave it for future work when they do.]
>
>
> ### Weakness 5
> > Do ablation of the hardcoded bits, i.e., you mention you hard-cache the first few and last few tokens. What is the contribution of this to your overall success metrics?
>
> We have conducted ablation studies allowing/disallowing sink tokens and recent tokens. H2O [1] (see Section 5.3 Q4) and Scissorhands [2] (see Section 4.1 "approach") also retain recent tokens sinks and determine these strategies are essential for full performance. We find a similar trend, as shown in the tables below. In fact, the cold-compress library turns this setting on by default due to the documented necessity of this strategy. Specifically, regardless of eviction strategy, the first 4 tokens of the prompt (the sinks according to [3]) are kept, and the 10 most recent tokens during every step of decoding are maintained.
>
> We believe this ablation study not only validates the necessity of maintaining these tokens for optimal performance but also aligns LSH-E’s configuration with standard practices in competing methods like H2O and Scissorhands. We hope that the ablation results strengthen the empirical foundation of our method, demonstrating that these design choices are essential and justified.
>
> ### Table: Ablation of Attention Sink Tokens and Recent Tokens on GSM8K Free Response Question Answering
> | Strategy | Cache Budget (%) | BertScore F1 | Rouge L | ChatGPT as a Judge Avg |
> |---|---|---|---|---|
> | LSH-E | 30% | 0.873 | 0.341 | 3.173 |
> | LSH-E no sink & recent | 30% | 0.652 | 0.048 | 1.028 |
> | L2 | 30% | 0.865 | 0.288 | 1.880 |
> | L2 no sink & recent | 30% | 0.844 | 0.228 | 1.270 |
> | LSH-E | 50% | 0.880 | 0.393 | 4.110 |
> | LSH-E no sink & recent | 50% | 0.777 | 0.173 | 1.513 |
> | L2 | 50% | 0.875 | 0.355 | 2.936 |
> | L2 no sink & recent | 50% | 0.866 | 0.318 | 2.217 |
> | LSH-E | 70% | 0.881 | 0.401 | 4.313 |
> | LSH-E no sink & recent | 70% | 0.841 | 0.295 | 2.687 |
> | L2 | 70% | 0.879 | 0.386 | 3.689 |
> | L2 no sink & recent | 70% | 0.876 | 0.374 | 3.390 |
> | LSH-E | 90% | 0.881 | 0.403 | 4.388 |
> | LSH-E no sink & recent | 90% | 0.868 | 0.363 | 3.630 |
> | L2 | 90% | 0.881 | 0.400 | 4.208 |
> | L2 no sink & recent | 90% | 0.880 | 0.397 | 4.110 |
>
> ### Weakness 6
> > The eviction policy is not clearly understandable in how it aggregates the hamming distances over time steps. Is it only based on the most recent time step, or some more complex rule?
>
> The Hamming distance is calculated per decoded token so it is based on the most recent time step and not aggregated over time steps.
>
> ### Question 1
>
> > Line 52: "However, this L2 dropout strategy only performs well on long-context retrieval tasks. It is specialized to retain only those tokens with the highest attention" -- be more specific. Why is this?
>
> We have updated our work to make this clearer. The L2 eviction strategy [6] was developed based on an empirical observation that smaller norm of key embedding correlates with higher attention score. For long-context retrieval tasks such Common Words, Needle-in-a-Haystack, etc., high-attention score tokens are the most important tokens since the question's text will overlap with the piece of context that needs to be retrieved. However, for generative tasks such as summarization, free response question-answering, etc., more than just high-attention tokens are required, which is why our method tends to outperform L2 on these benchmarks for most compression settings.
>
> ### Question 2
> > Line 57: "wide variety of tasks?" -- how do you define this?
>
> The common task types for KV cache compression experiments include multiple-choice, free response question-answering, long-context retrieval, and summarization. In our original draft, we included two benchmarks for each task type except for summarization -- which we have now added: MultiNews and GovReport from LongBench [4].

---

> ### Author Response · Authors · 2024-11-28
> **Rebuttal by Authors Pt 3**
>
> ### Question 3
> > Line 145: Formally for our setup, distd(x, y) cos θx,y, here it is more a measure of cosine similarity than distance. Misleading, perhaps?
>
> Since LSH involves transferring a similarity measure in a higher-dimensional space (in our case, cosine similarity), to a similarity measure in a lower-dimensional space (in our case, Hamming distance), we used the notation $dist$ for notational convenience. We have clarified this and also emphasized we are not referring to cosine distance.
>
> ### Question 4
> > Line 419: did you mean "LSH dimension does significantly impact performance" --> does not?
>
> Thank you for pointing this out. It was a typo and we mean "does not". We have fixed this error in the paper.
>
> ---
> Thank you for your review. If we have addressed your questions, we would appreciate it if you would consider updating your score. If any other questions or concerns remain, please let us know.
>
> ## References
>
> [1] Zhang, Z., Sheng, Y., Zhou, T., Chen, T., Zheng, L., Cai, R., ... & Chen, B. (2023). H2o: Heavy-hitter oracle for efficient generative inference [...].
>
> [2] Liu, Z., Desai, A., Liao, F., Wang, W., Xie, V., Xu, Z., ... & Shrivastava, A. (2024). Scissorhands: Exploiting the persistence [...]
>
> [3] Xiao, G., Tian, Y., Chen, B., Han, S., & Lewis, M. (2023). Efficient streaming language models with attention sinks.
>
> [4] Bai, Y., Lv, X., Zhang, J., Lyu, H., Tang, J., Huang, Z., ... & Li, J. (2023). Longbench: A bilingual, multitask benchmark for long context understanding.
>
> [5] Ge, S., Zhang, Y., Liu, L., Zhang, M., Han, J., & Gao, J. (2023). Model tells you what to discard: Adaptive kv cache compression for llms.
>
> [6] Devoto, A., Zhao, Y., Scardapane, S., & Minervini, P. (2024). A Simple and Effective $ L_2 $ Norm-Based Strategy for KV Cache Compression. arXiv preprint arXiv:2406.11430.
>
> [7] Bai, Y., Lv, X., Zhang, J., Lyu, H., Tang, J., Huang, Z., Du, Z., Liu, X., Zeng, A., Hou, L. and Dong, Y. (2023). Longbench: A bilingual, multitask benchmark for long context understanding.
>
> [8] Charikar, M. S. (2002, May). Similarity estimation techniques from rounding algorithms.
>
> [9] Kitaev, N., Kaiser, Ł., & Levskaya, A. (2020). Reformer: The efficient transformer.

---

> ### Author Response · Authors · 2024-12-01
>
> Dear Reviewer,
>
> We greatly appreciate your feedback. We have addressed your questions and concerns in our rebuttal. Please let us know if you have any further comments.
>
> Thank you.

---

### Official Review · Reviewer_R9hV · 2024-11-12

**Soundness:** 2
**Presentation:** 2
**Contribution:** 1
**Rating:** 3
**Confidence:** 4

**Summary:**

The idea is to reduce KV Cache by evicting and permanently dropping tokens at each position in the query. The heuristic used is to evict the lowest attention scored keys ( which is essentially similar to H2O / Scissorhands which preserve top attention scored keys). The difference is to use LSH to do a approximate score ranking to avoid SoftMax for exact computation.

**Strengths:**

Uses LSH to approximate attention computation for eviction (if you compare to h2o / scissorhands)

**Weaknesses:**

- Novelty: The novelty is limited.
- H2O / Scissorhands are known to not perform well on longbenchmark. Can we see some results on longbenchmark like passage retrieval datasets ?
- Missing baselines --only baseline used is L2 norm.
- Limited evaluation. can we get more results on longbenchmark at different budgets with standard baselines.

**Questions:**

see questions above,

---

> ### Author Response · Authors · 2024-11-28
> **Rebuttal by Authors Pt 1**
>
> We thank the reviewer for the valuable feedback and suggestions. We appreciate your recognition that our application of LSH approximate attention computation for eviction is efficient and a strength. Below, we address all stated weaknesses and questions.
>
> ### Weakness 1
> > Novelty: The novelty is limited.
>
> Could the reviewer expand on this? We are novel in several ways:
>
>  1. To the best of our knowledge, we are the only work using LSH for token eviction. Other works such as Reformer, QJL, Hyperattention, Subgen, and KDEFormer [5-9] use LSH to accelerate the attention computation but must initially view all queries, keys, and possibly the entire attention matrix, risking VRAM blowup.
>
>  2. We are the only attention-free token eviction strategy that makes a probabilistically-guaranteed estimation of attention (via known statistical properties of LSH). The only other comparable strategy, L2 eviction, relies on an observed correlation (low L2 norm = high attention), which may not hold for all transformer-based models and layers (see Figure 7 in our paper).
>
>  3. We propose a strategy which does not require construction of query/key embeddings of the entire context. We acquire the attention embeddings only for a percentage of the context that can fit within the user's memory budget and then perform token-by-token eviction for the remainder of the context. Interestingly, this approach does not appear in existing KV cache literature and we've integrated it for all other baselines. This may be regarded to an alternative strategy to contextual chunking.
>
> ### Weakness 2
>
> > H2O / Scissorhands are known to not perform well on longbenchmark. Can we see some results on longbenchmark like passage retrieval datasets ?
>
> > Missing baselines --only baseline used is L2 norm. - Limited evaluation. can we get more results on longbenchmark at different budgets with standard baselines.
>
> Thanks for this suggestion. We expanded the experiments to include two new tasks from the LongBench benchmarks: MultiNews and GovReport. Both are long-context summarization tasks. Both are long-context summarization tasks, since this task type was missing from our suite of evaluations.
>
> Additionally, we added comparisons to well-cited KV cache compression strategies, such as H2O [1], Scissorhands [2], and FastGen [5]. We have updated existing experiments in the paper to include these new baselines. We also provide results of the two summarization tasks in tables below.
>
> In these new experiments, LSH-E consistently demonstrats comparable or superior Rouge L scores across various cache budgets. In the MultiNews summarization task, LSH-E achieves higher Rouge L score at most cache budgets, outperforming all baselines, demonstrating LSH-E’s robustness and effectiveness in handling very large context lengths.
>
> We also measured throughput metrics on the MultiNews summarization task.  Per the throughput table below, our method performs better than the baselines on these two tasks across multiple KV cache budgets and our pre-fill speed is **1.5-2x as fast** as attention-dependent methods like H2O and Scissorhands, and even faster compared to FastGen.
>
> ### Table: Results of LongBench GovReport and MultiNews Summarization with Throughput
> |  |  | GovReport | MultiNews |  |  |
> |---|---|---|---|---|---|
> | Strategy | Cache Budget | Rouge L | Rouge L | Decode Toks Per Sec | Prefill Toks Per Sec |
> | Full | 100% | 0.230 | 0.192 | 16.071 | 16573.492 |
> | LSH-E | 30% | 0.202 | 0.180 | 22.880 | 20293.524 |
> | L2 | 30% | 0.201 | 0.165 | 23.981 | 20628.160 |
> | H2O | 30% | 0.219 | 0.175 | 21.555 | 13025.776 |
> | Scissorhands | 30% | 0.214 | 0.175 | 21.448 | 13004.254 |
> | LSH-E | 50% | 0.217 | 0.186 | 22.846 | 20459.961 |
> | L2 | 50% | 0.214 | 0.174 | 16.013 | 15851.952 |
> | H2O | 50% | 0.225 | 0.181 | 21.973 | 13969.985 |
> | Scissorhands | 50% | 0.219 | 0.182 | 20.978 | 13549.967 |
> | LSH-E | 70% | 0.223 | 0.187 | 22.914 | 21002.334 |
> | L2 | 70% | 0.223 | 0.187 | 24.305 | 21303.763 |
> | H2O | 70% | 0.229 | 0.184 | 21.793 | 14050.521 |
> | Scissorhands | 70% | 0.226 | 0.183 | 21.705 | 13954.693 |
> | LSH-E | 90% | 0.228 | 0.185 | 22.873 | 21229.230 |
> | L2 | 90% | 0.230 | 0.186 | 24.010 | 21305.693 |
> | H2O | 90% | 0.227 | 0.181 | 21.665 | 14007.697 |
> | Scissorhands | 90% | 0.230 | 0.182 | 21.411 | 14025.440 |
> | Fastgen | Attention recovery frac 70% | 0.192 | 0.129 | 12.752 | 1171.069 |
> | Fastgen | Attention recovery frac 75% | 0.231 | 0.174 | 12.291 | 1157.987 |
> | Fastgen | Attention recovery frac 80% | 0.232 | 0.184 | 11.850 | 1142.679 |
> | Fastgen | Attention recovery frac 85% | 0.236 | 0.183 | 11.658 | 1164.689 |

---

> ### Author Response · Authors · 2024-11-28
> **Rebuttal by Authors Pt 2**
>
> ## References
>
> [1] Zhang, Z., Sheng, Y., Zhou, T., Chen, T., Zheng, L., Cai, R., ... & Chen, B. (2023). H2o: Heavy-hitter oracle for efficient generative inference [...].
>
> [2] Liu, Z., Desai, A., Liao, F., Wang, W., Xie, V., Xu, Z., ... & Shrivastava, A. (2024). Scissorhands: Exploiting the persistence [...]
>
> [3] Bai, Y., Lv, X., Zhang, J., Lyu, H., Tang, J., Huang, Z., ... & Li, J. (2023). Longbench: A bilingual, multitask benchmark for long context understanding.
>
> [4] Ge, S., Zhang, Y., Liu, L., Zhang, M., Han, J., & Gao, J. (2023). Model tells you what to discard: Adaptive kv cache compression for llms.
>
> [5] Kitaev, N., Kaiser, Ł., & Levskaya, A. (2020). Reformer: The efficient transformer.
>
> [6] Zandieh, A., Daliri, M., & Han, I. (2024). QJL: 1-Bit Quantized JL Transform for KV Cache Quantization with Zero Overhead.
>
> [7] Han, I., Jayaram, R., Karbasi, A., Mirrokni, V., Woodruff, D. P., & Zandieh, A. (2023). Hyperattention: Long-context attention in near-linear time.
>
> [8] Zandieh, A., Han, I., Daliri, M., & Karbasi, A. (2023, July). Kdeformer: Accelerating transformers via kernel density estimation.
>
> [9] Zandieh, A., Han, I., Mirrokni, V., & Karbasi, A. (2024). SubGen: Token Generation in Sublinear Time and Memory.

---

> ### Author Response · Authors · 2024-12-01
>
> Dear Reviewer,
>
> We greatly appreciate your feedback. We have addressed your questions and concerns in our rebuttal. Please let us know if you have any further comments.
>
> Thank you,

---

### Official Review · Reviewer_HtGz · 2024-11-23

**Soundness:** 2
**Presentation:** 2
**Contribution:** 2
**Rating:** 5
**Confidence:** 4

**Summary:**

LLMs utilize KV cache to accelerate inference but take up significant GPU memory. LSH-E is an algorithm that uses LSH to compress the KV cache by evicting tokens that are cosine dissimilar. The token eviction happens pre-attention, thus making this method computationally affordable.

**Strengths:**

1. The small size of the KV cache allows it to be stored in GPU memory, eliminating latency from moving data between CPU and GPU.
2. KV cache eviction happens before attention computation, cutting down on unnecessary and expensive attention computations.
3. The greedy eviction approach makes it computationally very affordable.

**Weaknesses:**

1. It would be helpful to have an ablation study of LSH-E's performance with different numbers of first and recent tokens cached.
2. The benchmarks seem limited; there are only two datasets per task and the improvement over the baseline is not very significant in Needle-in-a-Haystack, Common Words, and MedQA Multiple Choice.
3. Evaluation does not include end-to-end speedup numbers, making it more difficult to see the ultimate impact of the contribution.
4. The greedy eviction algorithm assumes that the attention score between a particular key vector and the current query vector is representative of the attention score with subsequent query vectors. While there is ample empirical exploration on the correlation between attention and inverted LSH hamming distance, I could not find provable theoretical guarantees about the quality of the KV cache under this greedy eviction strategy or empirical observations about the consistency of attention scores across query vectors that suggest the soundness of this assumption. This is in contrast to other greedy approaches such as H2O that uses *accumulated* attention to be more robust to variations between individual query tokens.

**Questions:**

1. Under "Configuration and Setup", it is mentioned that you "keep the most recent 10 tokens and the first 4 tokens of the prompt always in the KV cache." Is the L2 eviction baseline also configured this way?
2. How well does LSH-E perform without keeping the most recent 10 tokens and the first 4 tokens?
3. Is it possible to perform more evaluations on LongBench tasks?
4. Do you have empirical results that show that the attention score for the current token is a reasonable proxy for attention scores for subsequent token, or that a low attention score for a current query token implies that the key token will not be critical to subsequent query tokens?

---

> ### Author Response · Authors · 2024-11-28
> **Rebuttal by Authors Pt 1**
>
> We thank the reviewer for the valuable feedback and suggestions. We are encouraged to see reviewer HtGz finds our approach computationally very affordable and appreciates our elimination of data transfer between the CPU and GPU. Below, we address all stated weaknesses and questions.
>
> ### Weakness 1 & Question 1
> > Under "Configuration and Setup", it is mentioned that you "keep the most recent 10 tokens and the first 4 tokens of the prompt always in the KV cache." Is the L2 eviction baseline also configured this way?
>
> Yes all other baselines are also configured in the same way.
>
> > It would be helpful to have an ablation study of LSH-E's performance with different numbers of first and recent tokens cached.
>
> We have conducted ablation studies allowing/disallowing sink tokens and recent tokens. H$_2$O [1] (see Section 5.3 Q4) and Scissorhands [2] (see Section 4.1 "approach") also retain recent tokens sinks and determine these strategies are essential for full performance. We find a similar trend, as shown in the tables below. In fact, the cold-compress library turns this setting on by default due to the documented necessity of this strategy. Specifically, regardless of eviction strategy, the first 4 tokens of the prompt (the sinks according to [3]) are kept, and the 10 most recent tokens during every step of decoding are maintained.
>
> We believe this ablation study not only validates the necessity of maintaining these tokens for optimal performance but also aligns LSH-E’s configuration with standard practices in competing methods like H2O and Scissorhands. We hope that the ablation results strengthen the empirical foundation of our method, demonstrating that these design choices are essential and justified.
>
> #### Ablation of Attention Sink Tokens and Recent Tokens on GSM8K Free Response Question Answering
> | Strategy | Cache Budget (%) | BertScore F1 | Rouge L | ChatGPT as a Judge Avg |
> |---|---|---|---|---|
> | LSH-E | 30% | 0.873 | 0.341 | 3.173 |
> | LSH-E no sink & recent | 30% | 0.652 | 0.048 | 1.028 |
> | L2 | 30% | 0.865 | 0.288 | 1.880 |
> | L2 no sink & recent | 30% | 0.844 | 0.228 | 1.270 |
> | LSH-E | 50% | 0.880 | 0.393 | 4.110 |
> | LSH-E no sink & recent | 50% | 0.777 | 0.173 | 1.513 |
> | L2 | 50% | 0.875 | 0.355 | 2.936 |
> | L2 no sink & recent | 50% | 0.866 | 0.318 | 2.217 |
> | LSH-E | 70% | 0.881 | 0.401 | 4.313 |
> | LSH-E no sink & recent | 70% | 0.841 | 0.295 | 2.687 |
> | L2 | 70% | 0.879 | 0.386 | 3.689 |
> | L2 no sink & recent | 70% | 0.876 | 0.374 | 3.390 |
> | LSH-E | 90% | 0.881 | 0.403 | 4.388 |
> | LSH-E no sink & recent | 90% | 0.868 | 0.363 | 3.630 |
> | L2 | 90% | 0.881 | 0.400 | 4.208 |
> | L2 no sink & recent | 90% | 0.880 | 0.397 | 4.110 |
>
> ### Weakness 2 , 3 & Question 3
>
> >The benchmarks seem limited; there are only two datasets per task [..].
>
> > Evaluation does not include end-to-end speedup numbers [...]
>
> > Is it possible to perform more evaluations on LongBench tasks?
>
> Thanks for this suggestion. We have added two LongBench [1] summarization tasks: MultiNews and GovReport. Additionally, we have added several other well-cited KV cache compression strategies: FastGen [2], H$_2$O [2], and ScissorHands [3]. We have updated existing experiments in the paper to include these new baselines. We also provide results of the two summarization tasks below.
>
> In these new experiments, LSH-E consistently demonstrats comparable or superior Rouge L scores across various cache budgets. In the MultiNews summarization task, LSH-E achieves higher Rouge L score at most cache budgets, outperforming all baselines, demonstrating LSH-E’s robustness and effectiveness in handling very large context lengths.
>
> We also report decoding and pre-fill tokens per second results on the LongBench MultiNews task. LSH-E is 1.5-2x as fast as H2O and Scissorhands, and 17x as fast as FastGen at the pre-fill stage. Even without low-level optimizations (e.g., expressing hash tables in binary bits), LSH-E proved to be as fast as the L2 strategy in decoding and significantly faster than attention-based baselines. This speedup was achieved while maintaining competitive quality metrics, demonstrating the computational efficiency of LSH-E. Please see the table below for details.

---

> ### Author Response · Authors · 2024-11-28
> **Rebuttal by Authors Pt 2**
>
> ### Table: Results of LongBench GovReport and MultiNews Summarization with Throughput
> |  |  | GovReport | MultiNews |  |  |
> |---|---|---|---|---|---|
> | Strategy | Cache Budget | Rouge L | Rouge L | Decode Toks Per Sec | Prefill Toks Per Sec |
> | Full | 100% | 0.230 | 0.192 | 16.071 | 16573.492 |
> | LSH-E | 30% | 0.202 | 0.180 | 22.880 | 20293.524 |
> | L2 | 30% | 0.201 | 0.165 | 23.981 | 20628.160 |
> | H2O | 30% | 0.219 | 0.175 | 21.555 | 13025.776 |
> | Scissorhands | 30% | 0.214 | 0.175 | 21.448 | 13004.254 |
> | LSH-E | 50% | 0.217 | 0.186 | 22.846 | 20459.961 |
> | L2 | 50% | 0.214 | 0.174 | 16.013 | 15851.952 |
> | H2O | 50% | 0.225 | 0.181 | 21.973 | 13969.985 |
> | Scissorhands | 50% | 0.219 | 0.182 | 20.978 | 13549.967 |
> | LSH-E | 70% | 0.223 | 0.187 | 22.914 | 21002.334 |
> | L2 | 70% | 0.223 | 0.187 | 24.305 | 21303.763 |
> | H2O | 70% | 0.229 | 0.184 | 21.793 | 14050.521 |
> | Scissorhands | 70% | 0.226 | 0.183 | 21.705 | 13954.693 |
> | LSH-E | 90% | 0.228 | 0.185 | 22.873 | 21229.230 |
> | L2 | 90% | 0.230 | 0.186 | 24.010 | 21305.693 |
> | H2O | 90% | 0.227 | 0.181 | 21.665 | 14007.697 |
> | Scissorhands | 90% | 0.230 | 0.182 | 21.411 | 14025.440 |
> | Fastgen | Attention recovery frac 70% | 0.192 | 0.129 | 12.752 | 1171.069 |
> | Fastgen | Attention recovery frac 75% | 0.231 | 0.174 | 12.291 | 1157.987 |
> | Fastgen | Attention recovery frac 80% | 0.232 | 0.184 | 11.850 | 1142.679 |
> | Fastgen | Attention recovery frac 85% | 0.236 | 0.183 | 11.658 | 1164.689 |
>
> ### Weakness 2
>
> >[...] the improvement over the baseline is not very significant in Needle-in-a-Haystack, Common Words, and MedQA Multiple Choice
>
> We respectfully disagree that our improvement is not significant. We strongly believe our approach overall is a useful addition to the toolkit of compression strategies. Per the throughput table below, we are 1.5-2x faster than H$_2$O, Scissorhands, and FastGen on pre-fill processing (resulting in thousands more tokens per second) while comparable in quality metrics. We are better than all methods on MedQA question-answering and LongBench MultiNews. Compared to L2 [6], our GPT-Judge scores are noticeably higher on MedQA and GSM-8K question-answering from 0.3 - 0.9 compression in all categories (by >1 point in several cases), indicating richer responses than L2 for generative language tasks.
>
> We also remind the reviewer the L2 strategy was originally designed for long-context retrieval tasks, and we are competitive against it down to 0.3 compression (at which point both methods significantly degrade). Our method defeats it at all compression rates on the LongBench MultiNews task as well. In summary, both of these zero-attention strategies, given their speed, are valuable strategies, with LSH-E preferable for text-generation tasks.
>
> ### Weakness 4 & Question 4
>
> >I could not find provable theoretical guarantees about the quality of the KV cache under this greedy eviction strategy or empirical observations [...]
>
> We measured and report the average atttention loss of the attention heads for both LSH-E, L2 and Scissorhands as empirical observation. Attention loss is defined as the sum of the attention probabilities for evicted tokens. Or equivalently, 1 - the sum of the attention probabilities for the tokens in the compressed cache.
>
> The attention loss was measured at 50% cache budget using prompts from the GSM8K question answering dataset. As per the table below, all three methods have low attention loss at 50% cache budget, and LSH-E has lower attention loss compared to L2 and Scissorhands, proving LSH-E's ability of keeping high attention tokens in the KV cache.
>
> ### **Table: Attention Loss**
> | Strategy| Attention Loss |
> |-----|------|
> | LSH-E | 0.03357896805 |
> | L2 | 0.03403072357 |
> | Scissorhands | 0.04483547211 |
>
> ----
> Thank you for your review. If we have addressed your questions, we would appreciate it if you would consider updating your score. If any other questions or concerns remain, please let us know.
>
> ### References
>
> [1] Zhang, Z., Sheng, Y., Zhou, T., Chen, T., Zheng, L., Cai, R., ... & Chen, B. (2023). H2o: Heavy-hitter oracle for efficient generative inference [...].
>
> [2] Liu, Z., Desai, A., Liao, F., Wang, W., Xie, V., Xu, Z., ... & Shrivastava, A. (2024). Scissorhands: Exploiting the persistence [...]
>
> [3] Xiao, G., Tian, Y., Chen, B., Han, S., & Lewis, M. (2023). Efficient streaming language models with attention sinks.
>
> [4] Bai, Y., Lv, X., Zhang, J., Lyu, H., Tang, J., Huang, Z., ... & Li, J. (2023). Longbench: A bilingual, multitask benchmark for long context understanding.
>
> [5] Ge, S., Zhang, Y., Liu, L., Zhang, M., Han, J., & Gao, J. (2023). Model tells you what to discard: Adaptive kv cache compression for llms.
>
> [6] Devoto, A., Zhao, Y., Scardapane, S., & Minervini, P. (2024). A Simple and Effective $ L_2 $ Norm-Based Strategy for KV Cache Compression. arXiv preprint arXiv:2406.11430.

---

> ### Author Response · Authors · 2024-12-01
>
> Dear Reviewer,
>
> We greatly appreciate your feedback. We have addressed your questions and concerns in our rebuttal. Please let us know if you have any further comments.
>
> Thank you,

---

> ### Comment · Reviewer_HtGz · 2024-12-02
> **Response to rebuttals**
>
> I thank the authors for a thorough response, especially the new benchmarks and ablation studies. I have a few more suggestions and thoughts:
> 1. The strength of LSH-E is that it pushes the Pareto frontier for the quality-throughput tradeoff. I suggest displaying a plot that clearly illustrates this.
> 2. Please include the sink-recent token ablation results in the paper. It would also be helpful to see "full attention" and "only sink and recent tokens" rows, as well as separate "with sink" and "with recent tokens" variants of L2 and LSH-E.
> 3. While the empirical results are intriguing, it still concerns me that there is no theoretical or intuitive explanation why the greedy approach is expected to work. This is even more concerning after seeing that LSH-E no sink performs very poorly but LSH-E with sink performs much better than L2. Maybe it would be help to show the individual GPT judging criteria (coherence, faithfulness, helpfulness) and examples that demonstrate that the individual criteria scores are reasonable. Perhaps this can provide some insights as to why the greedy approach works. For example, maybe the greedy approach leads to the premature eviction of most recent tokens while forcing the data structure to keep these tokens for a few iterations has a regularizing effect.

---

> ### Author Response · Authors · 2024-12-03
> **Response from authors pt1**
>
> We would like to thank the reviewer for the constructive feedback. Below, we address all stated suggestions and questions.
>
> > The strength of LSH-E is that it pushes the Pareto frontier for the quality-throughput tradeoff. I suggest displaying a plot that clearly illustrates this.
>
> Thank you for recognizing the strength of LSH-E. We will include such a plot in the next revision of the paper.
>
> > Please include the sink-recent token ablation results in the paper. It would also be helpful to see "full attention" and "only sink and recent tokens" rows, as well as separate "with sink" and "with recent tokens" variants of L2 and LSH-E.
>
> Thanks for the suggestion. We performed additional ablations including using only sink and recent tokens as a strategy, and L2 and LSH-E with only sink tokens, and with only recent tokens. In this ablation the LSH dimension was set to 16 bits. The number of sink tokens is 4 and the number of recent tokens is 10 except for the pure Sink & Recent strategy, which keeps (cache_size - 4) most recent tokens. Please see the updated table below for details. We will include the results of the sink-recent ablation in the next revision of the paper.
>
> ### Table: Ablation of Attention Sink Tokens and Recent Tokens on GSM8K Free Response Question Answering
> | Cache Budget | Strategy | Bert F1 | Rouge L | GPT Rouge | GPT Coherence | GPT Faithfulness | GPT Helpfulness |
> |---|---|---|---|---|---|---|---|
> | 10% | LSH-E | 0.831 | 0.157 | 1.018 | 1.387 | 1.147 | 1.083 |
> | 10% | LSH-E no sink no recent | 0.708 | 0.025 | 1.000 | 1.000 | 1.000 | 1.000 |
> | 10% | LSH-E no sink | 0.713 | 0.027 | 1.000 | 1.000 | 1.000 | 1.000 |
> | 10% | LSH-E no recent | 0.847 | 0.189 | 1.100 | 2.002 | 1.348 | 1.326 |
> | 10% | L2 | 0.826 | 0.151 | 1.005 | 1.293 | 1.098 | 1.033 |
> | 10% | L2 no sink no recent | 0.804 | 0.130 | 1.000 | 1.088 | 1.030 | 1.016 |
> | 10% | L2 no sink | 0.836 | 0.178 | 1.026 | 1.600 | 1.138 | 1.096 |
> | 10% | L2 no recent | 0.829 | 0.171 | 1.014 | 1.394 | 1.098 | 1.032 |
> | 10% | Sink & Recent | 0.843 | 0.176 | 1.040 | 1.882 | 1.298 | 1.248 |
> | 30% | LSH-E | 0.873 | 0.341 | 2.520 | 3.767 | 3.216 | 3.190 |
> | 30% | LSH-E no sink no recent | 0.744 | 0.068 | 1.004 | 1.024 | 1.018 | 1.006 |
> | 30% | LSH-E no sink | 0.744 | 0.066 | 1.006 | 1.018 | 1.028 | 1.002 |
> | 30% | LSH-E no recent | 0.873 | 0.342 | 2.546 | 3.956 | 3.340 | 3.472 |
> | 30% | L2 | 0.865 | 0.288 | 1.356 | 2.428 | 1.895 | 1.841 |
> | 30% | L2 no sink no recent | 0.844 | 0.228 | 1.040 | 1.478 | 1.292 | 1.268 |
> | 30% | L2 no sink | 0.865 | 0.290 | 1.474 | 2.750 | 2.010 | 2.102 |
> | 30% | L2 no recent | 0.846 | 0.238 | 1.032 | 1.478 | 1.320 | 1.272 |
> | 30% | Sink & Recent | 0.868 | 0.310 | 1.910 | 3.432 | 2.616 | 2.682 |
> | 50% | LSH-E | 0.880 | 0.393 | 3.457 | 4.530 | 4.212 | 4.241 |
> | 50% | LSH-E no sink no recent | 0.803 | 0.178 | 1.322 | 1.570 | 1.696 | 1.424 |
> | 50% | LSH-E no sink | 0.802 | 0.179 | 1.362 | 1.554 | 1.684 | 1.440 |
> | 50% | LSH-E no recent | 0.880 | 0.399 | 3.624 | 4.638 | 4.338 | 4.446 |
> | 50% | L2 | 0.875 | 0.355 | 2.190 | 3.494 | 3.035 | 3.027 |
> | 50% | L2 no sink no recent | 0.866 | 0.318 | 1.548 | 2.690 | 2.320 | 2.308 |
> | 50% | L2 no sink | 0.876 | 0.359 | 2.492 | 3.710 | 3.170 | 3.276 |
> | 50% | L2 no recent | 0.866 | 0.319 | 1.570 | 2.686 | 2.382 | 2.336 |
> | 50% | Sink & Recent | 0.879 | 0.385 | 3.412 | 4.488 | 4.054 | 4.122 |
> | 70% | LSH-E | 0.881 | 0.401 | 3.734 | 4.671 | 4.404 | 4.444 |
> | 70% | LSH-E no sink no recent | 0.847 | 0.295 | 2.350 | 2.818 | 2.912 | 2.612 |
> | 70% | LSH-E no sink | 0.847 | 0.295 | 2.332 | 2.794 | 2.888 | 2.600 |
> | 70% | LSH-E no recent | 0.881 | 0.402 | 3.884 | 4.790 | 4.546 | 4.650 |
> | 70% | L2 | 0.879 | 0.386 | 2.934 | 4.184 | 3.817 | 3.820 |
> | 70% | L2 no sink no recent | 0.876 | 0.374 | 2.684 | 3.836 | 3.510 | 3.528 |
> | 70% | L2 no sink | 0.879 | 0.390 | 3.266 | 4.370 | 4.018 | 4.104 |
> | 70% | L2 no recent | 0.876 | 0.374 | 2.718 | 3.842 | 3.522 | 3.516 |
> | 70% | Sink & Recent | 0.881 | 0.401 | 3.810 | 4.720 | 4.428 | 4.508 |
> | 90% | LSH-E | 0.881 | 0.403 | 3.837 | 4.722 | 4.468 | 4.525 |
> | 90% | LSH-E no sink no recent | 0.868 | 0.363 | 3.222 | 3.784 | 3.826 | 3.618 |
> | 90% | LSH-E no sink | 0.869 | 0.363 | 3.248 | 3.822 | 3.854 | 3.628 |
> | 90% | LSH-E no recent | 0.882 | 0.406 | 4.018 | 4.788 | 4.562 | 4.650 |
> | 90% | L2 | 0.881 | 0.400 | 3.569 | 4.578 | 4.324 | 4.361 |
> | 90% | L2 no sink no recent | 0.880 | 0.397 | 3.460 | 4.486 | 4.210 | 4.282 |
> | 90% | L2 no sink | 0.881 | 0.402 | 3.752 | 4.658 | 4.388 | 4.470 |
> | 90% | L2 no recent | 0.880 | 0.397 | 3.438 | 4.482 | 4.188 | 4.238 |
> | 90% | Sink & Recent | 0.881 | 0.405 | 4.006 | 4.792 | 4.572 | 4.644 |
> | 100% | Full | 0.882 | 0.403 | 3.845 | 4.716 | 4.499 | 4.545 |
>
> From the results we can see that sink tokens have a bigger impact on the performance of LSH while recent tokens impact L2 more.

---

> ### Author Response · Authors · 2024-12-03
> **Response from authors pt 2**
>
> > While the empirical results are intriguing, it still concerns me that there is no theoretical or intuitive explanation why the greedy approach is expected to work. This is even more concerning after seeing that LSH-E no sink performs very poorly but LSH-E with sink performs much better than L2.
> Maybe it would be help to show the individual GPT judging criteria (coherence, faithfulness, helpfulness) and examples that demonstrate that the individual criteria scores are reasonable. Perhaps this can provide some insights as to why the greedy approach works. For example, maybe the greedy approach leads to the premature eviction of most recent tokens while forcing the data structure to keep these tokens for a few iterations has a regularizing effect.
>
> In regards to variations in performance with and without the sink, we point the reviewer towards [1], which empirically examines that the sink registers significant attention regardless of layer, head, and decoding step. Methods such as H$_2$O and the cold-compress library retains the sink by default in response to this observation. Although the sink is important for high performance, in our opinion it is not related to the success of the greedy eviction approach.
>
>
> We provide an informal "proof sketch" on the error of LSH-E, that is, how much the compressed KV cache per our strategy deviated from the uncompressed cache. We leverage "The Persistence of Importance Hypothesis" first suggested and observed in the well-cited Scissorhands paper (Liu et al., 2023) [2]. Tokens which are "influential" at one timestep (i.e., produce a high attention score with the current token), tend to produce high attention for later steps. Interestingly, the authors, like us, use the inverse of the hypothesis to inform token dropping: tokens with low attention scores should be dropped as they will not be influential later.
>
> Assume the hypothesis is true and that our LSH attention estimation is exact.$^*$ Then Theorem 4.1 of [2] can be directly applied to LSH-E, which assumes a single token is dropped each timestep: *"Notice that when $m = 1$, i.e., in each iteration, we drop one token with the lowest score, the cache will always maintain $B$ tokens. If the ranking of the attention scores does not change in each iteration, Algorithm 2 will always drop tokens with the smallest attention scores."* Per this theorem, the upper bound on attention loss error of LSH-E scales directly with the imposed budget $B$, i.e., it decreases with larger budget.
>
> $^*$Our LSH estimation is not exact. The error is probabilistically controlled by the LSH dimension. Assuming a new, independently generated Gaussian projection is used at each timestep for the LSH, the probability of the LSH being correct is independent for each step, and thus multiplicative. Consequently, the user sets the sketch length sufficiently large to achieve a desired confidence $\delta$. Typical to sketching theory, the guarantees are typically far more aggressive than what is practically achievable: we use modest, fixed sketch dimension of 16 and do not refresh the Gaussian sketch/projection -- we simply maintain the existing hash codes in our dictionary and add new ones with the same sketch.
>
> Both the Scissorhands estimator and our LSH estimator are inexact (which both use restricted context windows per available memory), but the error in both cases seemingly does not significantly impact language output. Since Scissorhands fully computes attention scores over its window, it tends to survive quality at very high compression, while ours trades increased error at higher compression rates for dramatically improved throughput -- **our estimator is far faster.**
>
>
>
> ## Reference
> [1] Xiao, G., Tian, Y., Chen, B., Han, S., & Lewis, M. (2023). Efficient streaming language models with attention sinks. arXiv preprint arXiv:2309.17453.
>
> [2] Liu, Z., Desai, A., Liao, F., Wang, W., Xie, V., Xu, Z., ... & Shrivastava, A. (2024). Scissorhands: Exploiting the persistence of importance hypothesis for llm kv cache compression at test time. Advances in Neural Information Processing Systems, 36.

---

> > ### Comment · Reviewer_HtGz · 2024-12-03
> > **Response to authors**
> >
> > Thank you so much for the new ablation numbers! It is very interesting that it performs better without a recent tokens, and that just recent tokens and sink perform almost as well as LSH-E with sink.

---

> ### Author Response · Authors · 2024-12-03
>
> Thank you for your continued feedback and discussion! If you believe that we have addressed most of your concerns and questions (such as sink ablations and intuitive explanation of the greedy approach via the Scissorhands Importance Persistence Hypothesis), please do consider re-assessing your score, otherwise if you leave further remarks we can address them during the author response period.

---

### Author Response · Authors · 2024-11-28
**Summary of Rebuttal Pt 1**

# General Comments

Thank you to all reviewers for your thoughtful feedback and constructive suggestions. Your comments have been invaluable in helping us refine and strengthen our work. We are encouraged by the recognition of the computational efficiency and simplicity of our proposed LSH-E method, as well as its potential as a practical strategy for KV cache compression in resource-constrained scenarios.

# Reviewer Highlights

We would like to summarize the highlights that reviwers appreciated in our paper:

* Reviewer HtGz believes that our attention-free approach is "**computationally** very **affordable**" and "cuts down on unnecessary and expensive attention computations".
* Reviewer NvGH, R9hV and a2yh commend our novel use of "LSH to approximate attention computation".  NvGH comments that it "contributes to both the **effectiveness** and **scalability** of the proposed method". a2yh comments that "the motivations and reasons why LSH can produce a good performance are **well discussed**."
* Reviewr yqyi remarks that our approach is "**simple** yet **elegant**" and that we did "**good evaluations** on a range of use-cases".
* Reviewer rWSu notes that our approach is "**simple** and **clear** with **illustrative examples**".

# Summary of Changes

To address the feedbacks from the reviewer, we made the following improvements:

## 1. Additional Benchmarks and Baselines
Reviewers suggested that we should add tasks with even longer context length. In response, we expanded the experiments to include two new tasks from the LongBench benchmarks: MultiNews and GovReport. Both are long-context summarization tasks.

Additionally, comparisons to well-cited KV cache compression strategies, such as H2O, Scissorhands, and FastGen, were added to contextualize LSH-E's performance against state-of-the-art baselines. We have updated existing experiments in the paper to include these new baselines. We also provide results of the two summarization tasks in Table 1 below.

In these new experiments, LSH-E consistently demonstrats comparable or superior Rouge L scores across various cache budgets. In the MultiNews summarization task, LSH-E achieves higher Rouge L score at most cache budgets, outperforming all baselines, demonstrating LSH-E’s robustness and effectiveness in handling very large context lengths.

## 2. Throughput Analysis
Another addition to this rebuttal is the inclusion of throughput metrics. We provide decoding and prefill tokens per second results on the LongBench MultiNews task. LSH-E is 1.5-2x faster than H2O and Scissorhands, and 17x faster than FastGen at the prefill stage. Even without low-level optimizations (e.g., expressing hash tables in binary bits), LSH-E proved to be as fast as the L2 strategy in decoding and significantly faster than attention-based baselines.

This speedup was achieved while maintaining competitive quality metrics, demonstrating the computational efficiency of LSH-E. The throughput results address reviewer concerns about runtime metrics and substantiate the claimed computational benefits.

### Table 1: Results of LongBench MultiNews Summarization with Throughput Metrics
|  |  | GovReport | MultiNews |  |  |
|---|---|---|---|---|---|
| Strategy | Cache Budget | Rouge L | Rouge L | Decode Toks Per Sec | Prefill Toks Per Sec |
| Full | 100% | 0.230 | 0.192 | 16.071 | 16573.492 |
| LSH-E | 30% | 0.202 | 0.180 | 22.880 | 20293.524 |
| L2 | 30% | 0.201 | 0.165 | 23.981 | 20628.160 |
| H2O | 30% | 0.219 | 0.175 | 21.555 | 13025.776 |
| Scissorhands | 30% | 0.214 | 0.175 | 21.448 | 13004.254 |
| LSH-E | 50% | 0.217 | 0.186 | 22.846 | 20459.961 |
| L2 | 50% | 0.214 | 0.174 | 16.013 | 15851.952 |
| H2O | 50% | 0.225 | 0.181 | 21.973 | 13969.985 |
| Scissorhands | 50% | 0.219 | 0.182 | 20.978 | 13549.967 |
| LSH-E | 70% | 0.223 | 0.187 | 22.914 | 21002.334 |
| L2 | 70% | 0.223 | 0.187 | 24.305 | 21303.763 |
| H2O | 70% | 0.229 | 0.184 | 21.793 | 14050.521 |
| Scissorhands | 70% | 0.226 | 0.183 | 21.705 | 13954.693 |
| LSH-E | 90% | 0.228 | 0.185 | 22.873 | 21229.230 |
| L2 | 90% | 0.230 | 0.186 | 24.010 | 21305.693 |
| H2O | 90% | 0.227 | 0.181 | 21.665 | 14007.697 |
| Scissorhands | 90% | 0.230 | 0.182 | 21.411 | 14025.440 |
| Fastgen | Attention recovery frac 70% | 0.192 | 0.129 | 12.752 | 1171.069 |
| Fastgen | Attention recovery frac 75% | 0.231 | 0.174 | 12.291 | 1157.987 |
| Fastgen | Attention recovery frac 80% | 0.232 | 0.184 | 11.850 | 1142.679 |
| Fastgen | Attention recovery frac 85% | 0.236 | 0.183 | 11.658 | 1164.689 |

---

### Author Response · Authors · 2024-11-28
**Summary of Rebuttal Pt 2**

## 3. Ablation Studies on Attention Sink Tokens and Recent Tokens
To address concerns about hardcoding specific tokens for retention, we conducted ablation studies on the impact of retaining attention sink tokens (first 4 tokens) and recent tokens (last 10 tokens). The results revealed that disabling these features led to performance degradation. For example, at a 50% cache budget on GSM8K, LSH-E without sink and recent tokens scored a Rouge L of 0.173 compared to 0.393 with these features enabled.

This study not only validated the necessity of maintaining these tokens for optimal performance but also aligned LSH-E’s configuration with standard practices in competing methods like H2O and Scissorhands. We hope that the ablation results strengthened the empirical foundation of our method, demonstrating that these design choices are essential and justified.

### Table 2: Ablation of Attention Sink Tokens and Recent Tokens on GSM8K Free Response Question Answering
| Strategy | Cache Budget (%) | BertScore F1 | Rouge L | ChatGPT as a Judge Avg |
|---|---|---|---|---|
| LSH-E | 30% | 0.873 | 0.341 | 3.173 |
| LSH-E no sink & recent | 30% | 0.652 | 0.048 | 1.028 |
| L2 | 30% | 0.865 | 0.288 | 1.880 |
| L2 no sink & recent | 30% | 0.844 | 0.228 | 1.270 |
| LSH-E | 50% | 0.880 | 0.393 | 4.110 |
| LSH-E no sink & recent | 50% | 0.777 | 0.173 | 1.513 |
| L2 | 50% | 0.875 | 0.355 | 2.936 |
| L2 no sink & recent | 50% | 0.866 | 0.318 | 2.217 |
| LSH-E | 70% | 0.881 | 0.401 | 4.313 |
| LSH-E no sink & recent | 70% | 0.841 | 0.295 | 2.687 |
| L2 | 70% | 0.879 | 0.386 | 3.689 |
| L2 no sink & recent | 70% | 0.876 | 0.374 | 3.390 |
| LSH-E | 90% | 0.881 | 0.403 | 4.388 |
| LSH-E no sink & recent | 90% | 0.868 | 0.363 | 3.630 |
| L2 | 90% | 0.881 | 0.400 | 4.208 |
| L2 no sink & recent | 90% | 0.880 | 0.397 | 4.110 |

## 4. Attention Loss Analysis

We added an analysis of attention loss for LSH-E, L2 and Scissorhands, quantifying the discrepancy introduced by the eviction strategy compared to maintaining the full cache. We measured the atttention loss of each attention head and report the average. Attention loss is defined as the sum of the attention probabilities for evicted tokens. Or equivalently, 1 - the sum of the attention probabilities for the tokens in the compressed cache.

The attention loss was measured at 50% cache budget using prompts from the GSM8K question answering dataset. As per Table 5, all three methods have low attention loss at 50% cache budget, and LSH-E has lower attention loss compared to L2 and scissorhands, proving LSH-E's ability of keeping high attention tokens in the KV cache.

By quantifying attention loss, we demonstrated that LSH-E introduces minimal deviation from full-cache attention, addressing concerns about the theoretical guarantees of its quality.

### Table 3: Attention Loss
| Strategy       | Attention Loss    |
|----------------|-------------------|
| LSH-E          | 0.03357896805     |
| L2             | 0.03403072357     |
| Scissorhands   | 0.04483547211     |


## 5. Clarified Novelty and Conceptual Differences
We clarified the conceptual distinctions between LSH-E and related works such as Reformer, H2O, and SubGen and updated the related works section of our paper. While LSH-E uses LSH for token eviction, Reformer and similar methods use LSH to accelerate attention computation. This distinction underscores LSH-E’s novelty as a probabilistically guaranteed attention-free token eviction strategy, separating it from approaches like L2 eviction. Additionally, it does not require scanning the entirety of the context like existing approaches, which risks VRAM blowup.

This clarification strengthens the claim of LSH-E's novelty and highlights its practical advantages, particularly in memory-constrained scenarios.

## 6. Improved Presentation and Addressed Minor Issues

We addressed several presentation issues, including improving axis captions in figures and fixing typographical errors. These changes enhanced the paper’s readability. Moreover, we expanded the discussion of results, providing intuitive explanations for observed trends, such as why LSH-E outperforms Full in certain cases and why its performance degrades at lower cache budgets.

---

### Author Response · Authors · 2024-11-28
**Summary of Rebuttal Pt 3**

## 7. Updated Ablation Studies on Attention Sink Tokens and Recent Tokens

We performed additional ablations including using only sink and recent tokens as a strategy, and L2 and LSH-E with only sink tokens, and with only recent tokens. The LSH dimension was set to 16 bits. The number of sink tokens is 4 and the number of recent tokens is 10 except for the Sink & Recent strategy, which keeps (cache_size - 4) most recent tokens. From the results we found that sink tokens have a bigger impact on the performance of LSH while recent tokens impact L2 more. Please see updated results in Table 4 below.

#### Table 4: Ablation of Attention Sink Tokens and Recent Tokens on GSM8K Free Response Question Answering
| Cache Budget | Strategy | Bert F1 | Rouge L | GPT Rouge | GPT Coherence | GPT Faithfulness | GPT Helpfulness |
|---|---|---|---|---|---|---|---|
| 10% | LSH-E | 0.831 | 0.157 | 1.018 | 1.387 | 1.147 | 1.083 |
| 10% | LSH-E no sink no recent | 0.708 | 0.025 | 1.000 | 1.000 | 1.000 | 1.000 |
| 10% | LSH-E no sink | 0.713 | 0.027 | 1.000 | 1.000 | 1.000 | 1.000 |
| 10% | LSH-E no recent | 0.847 | 0.189 | 1.100 | 2.002 | 1.348 | 1.326 |
| 10% | L2 | 0.826 | 0.151 | 1.005 | 1.293 | 1.098 | 1.033 |
| 10% | L2 no sink no recent | 0.804 | 0.130 | 1.000 | 1.088 | 1.030 | 1.016 |
| 10% | L2 no sink | 0.836 | 0.178 | 1.026 | 1.600 | 1.138 | 1.096 |
| 10% | L2 no recent | 0.829 | 0.171 | 1.014 | 1.394 | 1.098 | 1.032 |
| 10% | Sink & Recent | 0.843 | 0.176 | 1.040 | 1.882 | 1.298 | 1.248 |
| 30% | LSH-E | 0.873 | 0.341 | 2.520 | 3.767 | 3.216 | 3.190 |
| 30% | LSH-E no sink no recent | 0.744 | 0.068 | 1.004 | 1.024 | 1.018 | 1.006 |
| 30% | LSH-E no sink | 0.744 | 0.066 | 1.006 | 1.018 | 1.028 | 1.002 |
| 30% | LSH-E no recent | 0.873 | 0.342 | 2.546 | 3.956 | 3.340 | 3.472 |
| 30% | L2 | 0.865 | 0.288 | 1.356 | 2.428 | 1.895 | 1.841 |
| 30% | L2 no sink no recent | 0.844 | 0.228 | 1.040 | 1.478 | 1.292 | 1.268 |
| 30% | L2 no sink | 0.865 | 0.290 | 1.474 | 2.750 | 2.010 | 2.102 |
| 30% | L2 no recent | 0.846 | 0.238 | 1.032 | 1.478 | 1.320 | 1.272 |
| 30% | Sink & Recent | 0.868 | 0.310 | 1.910 | 3.432 | 2.616 | 2.682 |
| 50% | LSH-E | 0.880 | 0.393 | 3.457 | 4.530 | 4.212 | 4.241 |
| 50% | LSH-E no sink no recent | 0.803 | 0.178 | 1.322 | 1.570 | 1.696 | 1.424 |
| 50% | LSH-E no sink | 0.802 | 0.179 | 1.362 | 1.554 | 1.684 | 1.440 |
| 50% | LSH-E no recent | 0.880 | 0.399 | 3.624 | 4.638 | 4.338 | 4.446 |
| 50% | L2 | 0.875 | 0.355 | 2.190 | 3.494 | 3.035 | 3.027 |
| 50% | L2 no sink no recent | 0.866 | 0.318 | 1.548 | 2.690 | 2.320 | 2.308 |
| 50% | L2 no sink | 0.876 | 0.359 | 2.492 | 3.710 | 3.170 | 3.276 |
| 50% | L2 no recent | 0.866 | 0.319 | 1.570 | 2.686 | 2.382 | 2.336 |
| 50% | Sink & Recent | 0.879 | 0.385 | 3.412 | 4.488 | 4.054 | 4.122 |
| 70% | LSH-E | 0.881 | 0.401 | 3.734 | 4.671 | 4.404 | 4.444 |
| 70% | LSH-E no sink no recent | 0.847 | 0.295 | 2.350 | 2.818 | 2.912 | 2.612 |
| 70% | LSH-E no sink | 0.847 | 0.295 | 2.332 | 2.794 | 2.888 | 2.600 |
| 70% | LSH-E no recent | 0.881 | 0.402 | 3.884 | 4.790 | 4.546 | 4.650 |
| 70% | L2 | 0.879 | 0.386 | 2.934 | 4.184 | 3.817 | 3.820 |
| 70% | L2 no sink no recent | 0.876 | 0.374 | 2.684 | 3.836 | 3.510 | 3.528 |
| 70% | L2 no sink | 0.879 | 0.390 | 3.266 | 4.370 | 4.018 | 4.104 |
| 70% | L2 no recent | 0.876 | 0.374 | 2.718 | 3.842 | 3.522 | 3.516 |
| 70% | Sink & Recent | 0.881 | 0.401 | 3.810 | 4.720 | 4.428 | 4.508 |
| 90% | LSH-E | 0.881 | 0.403 | 3.837 | 4.722 | 4.468 | 4.525 |
| 90% | LSH-E no sink no recent | 0.868 | 0.363 | 3.222 | 3.784 | 3.826 | 3.618 |
| 90% | LSH-E no sink | 0.869 | 0.363 | 3.248 | 3.822 | 3.854 | 3.628 |
| 90% | LSH-E no recent | 0.882 | 0.406 | 4.018 | 4.788 | 4.562 | 4.650 |
| 90% | L2 | 0.881 | 0.400 | 3.569 | 4.578 | 4.324 | 4.361 |
| 90% | L2 no sink no recent | 0.880 | 0.397 | 3.460 | 4.486 | 4.210 | 4.282 |
| 90% | L2 no sink | 0.881 | 0.402 | 3.752 | 4.658 | 4.388 | 4.470 |
| 90% | L2 no recent | 0.880 | 0.397 | 3.438 | 4.482 | 4.188 | 4.238 |
| 90% | Sink & Recent | 0.881 | 0.405 | 4.006 | 4.792 | 4.572 | 4.644 |
| 100% | Full | 0.882 | 0.403 | 3.845 | 4.716 | 4.499 | 4.545 |

---

### Author Response · Authors · 2024-11-28
**Summary of Rebuttal Pt 4**

# Final Comment
We greatly appreciate reviewer feedback. Our rebuttal addresses all questions and concerns. We would appreciate it if the reviewers could update their scores accordingly. Please let us know if you have more comments or questions.

# References

[1] Zhang, Z., Sheng, Y., Zhou, T., Chen, T., Zheng, L., Cai, R., ... & Chen, B. (2023). H2o: Heavy-hitter oracle for efficient generative inference of large language models.

[2] Liu, Z., Desai, A., Liao, F., Wang, W., Xie, V., Xu, Z., ... & Shrivastava, A. (2024). Scissorhands: Exploiting the persistence of importance hypothesis for llm kv cache compression at test time.

[3] Xiao, G., Tian, Y., Chen, B., Han, S., & Lewis, M. (2023). Efficient streaming language models with attention sinks.

[4] Bai, Y., Lv, X., Zhang, J., Lyu, H., Tang, J., Huang, Z., ... & Li, J. (2023). Longbench: A bilingual, multitask benchmark for long context understanding.

[5] Ge, S., Zhang, Y., Liu, L., Zhang, M., Han, J., & Gao, J. (2023). Model tells you what to discard: Adaptive kv cache compression for llms.

[6] Devoto, A., Zhao, Y., Scardapane, S., & Minervini, P. (2024). A Simple and Effective $ L_2 $ Norm-Based Strategy for KV Cache Compression. arXiv preprint arXiv:2406.11430.

[7] Bai, Y., Lv, X., Zhang, J., Lyu, H., Tang, J., Huang, Z., Du, Z., Liu, X., Zeng, A., Hou, L. and Dong, Y. (2023). Longbench: A bilingual, multitask benchmark for long context understanding.

[8] Charikar, M. S. (2002, May). Similarity estimation techniques from rounding algorithms.

[9] Kitaev, N., Kaiser, Ł., & Levskaya, A. (2020). Reformer: The efficient transformer.

---

### Note · Authors · 2025-01-22

I have read and agree with the venue's withdrawal policy on behalf of myself and my co-authors.